# Machine Learning Deciphers $CO_2$ Sequestration and Subsurface Flowpaths from Stream Chemistry

Andrew R. Shaughnessy[1], Xin Gu[2], Tao Wen[2,3], Susan L. Brantley[1,2]

1. Department of Geosciences, Pennsylvania State University, University Park, PA, USA
2. Earth and Environmental Systems Institute, Pennsylvania State University, University Park, PA, USA
3. Department of Earth and Environmental Science, Syracuse University, Syracuse, NY, USA

*Correspondence to*: Andrew R. Shaughnessy (ars637@psu.edu)

**Abstract.** Endmember mixing analysis (EMMA) is often used by hydrogeochemists to interpret the sources of stream solutes, but variations in stream concentrations and discharges remain difficult to explain. We discovered that machine learning can be used to highlight patterns in stream chemistry that reveal information about sources of solutes and subsurface groundwater flowpaths. The investigation has implications, in turn, for the balance of $CO_2$ in the atmosphere. For example, $CO_2$-driven weathering of silicate minerals removes carbon from the atmosphere over $\sim 10^6$-yr timescales. Weathering of another common mineral, pyrite, releases sulfuric acid that in turn causes dissolution of carbonates. In that process, however, $CO_2$ is released instead of sequestered from the atmosphere. Thus, to understand long-term global $CO_2$ sequestration by weathering requires quantification of $CO_2$- versus $H_2SO_4$- driven reactions. Most researchers estimate such weathering fluxes from stream chemistry but interpreting the reactant minerals and acids dissolved in streams has been fraught with difficulty. We apply a machine learning technique to EMMA in three watersheds to determine the extent of mineral dissolution by each acid, without pre-defining the endmembers. The results show that the watersheds continuously or intermittently sequester $CO_2$ but the extent of $CO_2$ drawdown is diminished in areas heavily affected by acid rain. Prior to applying the new algorithm, $CO_2$ drawdown was overestimated. The new technique, which elucidates the importance of different subsurface flowpaths and long-timescale changes in the watersheds, should have utility as a new EMMA for investigating water resources worldwide.

## 1 Introduction

We need to understand the long-term controls on atmospheric $CO_2$ because of the impact of this greenhouse gas on global climate. This is important because humans are increasingly burning fossil fuels and releasing long-sequestered carbon to the atmosphere (Kasting and Walker, 1992). This new C flux upsets the natural long-term balance in the atmosphere between volcanic degassing and weathering-induced drawdown of $CO_2$ over millennial timescales. Chemical weathering of the most common rock-forming minerals, silicates and carbonates, removes $CO_2$ from the atmosphere by forming dissolved inorganic carbon that is carried in rivers to the ocean (DIC; Fig. 1). Over $10^5 - 10^6$ yr timescales, this DIC is precipitated as marine calcite, releasing half or all of the atmospherically derived $CO_2$ back to the atmosphere for silicates and carbonates, respectively (Fig. 1). Thus, over this timescale, $CO_2$-driven weathering ($CO_2$-weathering) of silicates sequesters $CO_2$ out of the atmosphere while $CO_2$-weathering of carbonates neither removes nor releases $CO_2$ to the atmosphere (Fig. 1). Some researchers also emphasize that this simple picture

neglects weathering of another ubiquitous mineral, pyrite (Lerman et al., 2007). When pyrite weathers, it produces sulfuric acid that also dissolves silicates and carbonates, i.e., $H_2SO_4$-weathering. When DIC generated through $H_2SO_4$-weathering of carbonates is carried to the ocean, marine calcite precipitates and releases $CO_2$, increasing atmosphere concentrations (Spence and Telmer, 2005; Calmels et al., 2011; Torres et al., 2014; Kölling et al., 2019). Thus, determination of the weathering contributions of silicates, carbonates, and pyrite is essential toward understanding long-term dynamics of $CO_2$. In this paper we describe a powerful machine learning technique to interpret the sources of stream solutes to understand problems such as weathering. While we show the importance of applying this machine learning technique to the weathering question, we also emphasize how machine learning can teach hydrogeochemists about subsurface flow paths and other characteristics of stream systems.

The most common way hydrogeochemists interpret the fluxes of weathering are to investigate stream and river chemistry. Determining the endmembers for streams is important because streams integrate the byproducts of weathering reactions over drainage basins, allowing assessment of regional to global understanding of fluxes – but only if minerals weathered by different acid sources can be deconvoluted (Li et al., 2008; Calmels et al., 2011; Torres et al., 2016; Winnick et al., 2017; Burke et al., 2018; Killingsworth et al., 2018). In small-scale studies in the laboratory or soil profiles, mineral reactions can be documented, but this information cannot be scaled up easily (Navarre-Sitchler and Brantley 2007). Here we show that machine learning can decipher the balance of fluxes of $CO_2$- versus $H_2SO_4$-weathering as recorded in stream chemistry. We discovered that catchments partition water into subsurface flowpaths that can be i) deciphered with respect to the extent of pyrite, silicate, and carbonate weathering in different lithologies, and ii) interpreted with respect to whether weathering is driven by $CO_2$ or $H_2SO_4$. We emphasize the long-term effects (over $10^5$ -$10^6$ yr) on the $CO_2$ balance in the atmosphere.

Although geochemists commonly use stream chemistry to determine mineral sources of solutes via weathering reactions over large aerial extents (Gaillardet et al., 1999) and hydrologists commonly use endmember mixing analysis (EMMA) to determine the sources of solutes in a stream (Christophersen et al., 1990), stream datasets remain difficult to interpret because of spatial and temporal variations in endmember composition. For example, sulfur isotopes in stream solutes can distinguish pyrite-derived from rain-derived sulfate because pyrite typically is depleted in [34]S (Burke et al., 2018; Killingsworth et al., 2018). But this attribution is difficult, more expensive, and often ambiguous because pyrite $\delta^{34}S$ varies between formations (Gautier, 1986) or within a single catchment (Bailey et al., 2004). Likewise, inputs of sulfate to watersheds, such as acid rain, can swamp out the signal from mineral reactions, and can change significantly over time (e.g., because of changing acid rain deposition) (Lynch et al., 2000; Lehmann et al., 2007). These factors make it difficult to determine sources releasing sulfate to varying stream chemistries over time.

Several so-called "inverse models" have been used successfully to partition sulfate into endmember sources for streams and rivers. These include the two prominent modeling approaches by Torres et al. (2016) and Burke et al. (2018). However, because the chemistry of acid rain has varied over the past decades, utilizing the full range of rain chemistry in those models results in unrealistic contributions of acid rain (i.e., > 100%) or models that fail to converge. This is at least partly because the chemistry of acid rain has been so variable that it spans the entire measured range of stream samples. Additionally, utilizing the approach of Burke et al. (2018), based on the approach

of Gaillardet et al. (1999), requires a priori assignment of accurate endmember chemistries. Often, the researcher must rely on a few samples to characterize endmembers, resulting in large uncertainties in endmember chemistry and in source apportioning. Since the inception of EMMA, many researchers have aimed to improve analysis through a more accurate determination of unknown or under-constrained endmember chemistries (Hooper, 2003; Carrera et al., 2004; Valder et al., 2012). But these efforts all use some a priori determination of endmembers. Our machine learning model adds to the growing effort to improve EMMA by applying blind source separation. The machine-learning approach we describe here de-convolves sources of stream chemistry without pre-defining the endmembers. We demonstrate this first with a synthetic dataset and then with data from three well-studied watersheds with different characteristics. The new method discovers the endmember chemistries and, as a result, documents new findings of importance previously undiscovered with the other methods.

For the target watersheds, we focus first on Shale Hills, an acid rain-impacted shale watershed in central Pennsylvania, USA with extensive data for water/rock chemistry (Jin et al., 2010; Brantley et al., 2013a; Sullivan et al., 2016). This watershed allows the most complete understanding of solute sources. Although we do not show this here, if we use either of the two previously used models for source attribution, stream chemistry data for Shale Hills either does not separate acid rain and pyrite as a sulfate source (if we use the model of Torres et al., 2016) or yields a proportion for acid rain which is larger than 100% (if we use the model of Burke et al., 2018). As shown below, the Non-negative Matrix Factorization (NMF) model easily defines endmembers and proportions.

We then show the utility of the machine learning method for watersheds where less water/rock chemistry has been published: we investigate East River and Hubbard Brook catchments. Like Shale Hills, East River is shale-hosted, but it receives little acid rain (Winnick et al., 2017). In contrast, Hubbard Brook has been extensively impacted by acid rain but is underlain by glacial till over schist (Likens et al., 2002). In both cases, NMF successfully determines endmembers and source proportions.

| Reaction Number | CO$_2$-Weathering of Silicates | Net Effect on Atmospheric CO$_2$ (mol CO$_2$/mol Mineral) | Timescale (yr) |
|---|---|---|---|
| 1 | $CaSiO_{3(s)} + 2CO_{2(g)} + H_2O \rightarrow Ca^{2+}_{(aq)} + 2HCO^-_{3(aq)} + SiO_{2(aq)}$ <br> Silicate Dissolution → Riverine DIC | -2 CO$_2$ | $<10^4$ |
| 2 | $CaSiO_{3(s)} + 2CO_{2(g)} + H_2O \rightarrow CaCO_{3(s)} + CO_{2(g)} + H_2O + SiO_{2(aq)}$ <br> Silicate Dissolution → Marine Carbonate Precipitation | -1 CO$_2$ | $10^5$-$10^6$ |

CO$_2$ - Weathering of Carbonates

| Reaction Number | | Net Effect | Timescale |
|---|---|---|---|
| 3 | $CaCO_{3(s)} + CO_{2(g)} + H_2O \rightarrow Ca^{2+}_{(aq)} + 2HCO^-_{3(aq)}$ <br> Terrestrial Carbonate Dissolution → Riverine DIC | -1 CO$_2$ | $<10^4$ |
| 4 | $CaCO_{3(s)} + CO_{2(g)} + H_2O \rightarrow CaCO_{3(s)} + CO_{2(g)} + H_2O$ <br> Terrestrial Carbonate Dissolution → Marine Carbonate Precipitation | 0 CO$_2$ | $10^5$-$10^6$ |

H$_2$SO$_4$–Weathering of Silicates

| Reaction Number | | Net Effect | Timescale |
|---|---|---|---|
| 5 | $CaSiO_{3(s)} + \frac{1}{2}FeS_{2(s)} + \frac{15}{8}O_{2(g)} + \frac{3}{4}H_2O \rightarrow Ca^{2+}_{(aq)} + SO^{2-}_{4(aq)} + SiO_{2(aq)} + \frac{1}{2}Fe(OH)_{3(s)}$ <br> Silicate Dissolution → Riverine Sulfate | 0 CO$_2$ | $10^5$-$10^6$ |

H$_2$SO$_4$-Weathering of Carbonates

| Reaction Number | | Net Effect | Timescale |
|---|---|---|---|
| 6 | $CaCO_{3(s)} + \frac{1}{4}FeS_{2(s)} + \frac{15}{16}O_{2(g)} + \frac{7}{8}H_2O \rightarrow Ca^{2+}_{(aq)} + HCO^-_{3(aq)} + \frac{1}{2}SO^{2-}_{4(aq)} + \frac{1}{4}Fe(OH)_{3(s)}$ <br> Terrestrial Carbonate Dissolution → Riverine DIC | 0 CO$_2$ | $<10^4$ |
| 7 | $CaCO_{3(s)} + \frac{1}{4}FeS_{2(s)} + \frac{15}{16}O_{2(g)} + \frac{3}{8}H_2O \rightarrow \frac{1}{2}CaCO_{3(s)} + \frac{1}{2}CO_{2(g)} + \frac{1}{2}SO^{2-}_{4(aq)} + \frac{1}{2}Ca^{2+}_{(aq)} + \frac{1}{4}Fe(OH)_{3(s)}$ <br> Terrestrial Carbonate Dissolution → Marine Carbonate Precipitation | $+\frac{1}{2}$ CO$_2$ | $10^5$-$10^6$ |

Figure 1. Schematic summarizing the reactions, timescales, and net CO₂ release to or uptake from the atmosphere accompanying weathering of silicate and carbonate minerals. Uptake or release depends upon timescale, as shown, and as discussed in text. CaSiO₃ is used as a generic silicate mineral.

## 2 Methods

### 2.1 Study Sites

Where previous deconvolutions of stream chemistry into endmembers were generally based on assumptions of the chemistry of dissolving minerals alone, data for watersheds show that the flowpath of the water also affects this chemistry (e.g., Brantley et al 2017). We demonstrate this with data from three well-studied watersheds with different characteristics. We focus first on Shale Hills, a small (0.08 km$^2$), acid-rain impacted forested watershed underlain by Rose Hill shale located in central Pennsylvania, USA (Brantley et al., 2018). The Rose Hill Formation shale contains ~0.14 wt% S as pyrite (FeS$_2$) (Gu et al., 2020a).

We then show utility of the method in East River (shale-hosted but it receives little acid rain) and Hubbard Brook (extensively impacted by acid rain but is underlain by schist and glacial till) catchments. Specifically, East River is a large (85 km$^2$), mountainous watershed underlain by Mancos Shale that is located near Gothic, Colorado USA within the Gunnison River basin (Winnick et al., 2017). The Mancos is a black shale that contains ~1.6 wt% S

as pyrite (Wan et al., 2019). Both of these shale-hosted watersheds contain carbonate minerals that vary in composition and abundance in the subsurface. Lastly, Hubbard Brook (Nezat et al., 2004), located in the White Mountains of New Hampshire USA, consists of a series of nine small (0.14-0.77 km$^2$), forested watersheds underlain by Rangeley Formation metamorphosed shale and sandstone (schist) generally covered by glacial till derived mostly from the Kinsman granodiorite. The schist bedrock contains ~0.2-0.9 wt% S and till contains ~0.1-0.2 wt% S. Again, almost all S is present as iron sulfide (pyrite or pyrrhotite). Both bedrock and till are largely carbonate-free.

**2.2 Data Acquisition**

For Shale Hills, daily stream chemistry has been reported from 2008-2010 (Brantley et al., 2013b; Brantley et al., 2013c; Brantley et al., 2013d). Additional samples were measured in other time intervals for sulfur isotopes and alkalinity (Jin et al., 2014). All samples were filtered through a 0.45 μm Nylon filter and aliquots for cation analysis were acidified with nitric acid. Cations were measured on a Leeman Labs PS3000UV (Teledyne Leeman Labs, Hudson, NH) inductively coupled plasma–optical emission spectrometer (ICP-OES), and anions were measured on a Dionex Ion Chromatograph (Sunnyvale, CA). Alkalinity was measured by titration with 0.16 M H$_2$SO$_4$. Discharge data are available online (http://www.czo.psu.edu/data_time_series.html).

All published data from East River were used in analysis (Winnick et al., 2017), except for two samples with extremely high values of chloride (246 and 854 μM) because they differed significantly from the remaining sample chemistry (average Cl concentration = 21μM). Hubbard Brook weekly chemistry from 2000-2017 was downloaded for the sub-catchments (3, 6, 7, 8, 9) that were not experimentally manipulated (Bernhardt et al., 2019). Stream discharge data for each sub-catchment are from USDA Forest Service (USDA, 2019).

**2.3 Machine Learning Model**

To assign the proportion of sulfate in streams to sources, we first bootstrapped measurements to increase data volume and then used a method of blind source separation (Alexandrov and Vesselinov, 2014; Vesselinov et al., 2018) called non-negative matrix factorization (NMF). NMF is unique from previously used methods in that it allows calculation of endmember compositions and mixing proportions simultaneously and does not rely on measurements or assumptions of endmembers a priori (Fig. 2A; see SM section 1). Specifically, NMF decomposes the $n$ x $m$ matrix, $V$, into two matrices $W$ and $H$:

$$V = WH, \tag{1}$$

Here, cell entries of $V$ are molar solute concentration ratios, $[X]/[Y]$, for stream samples. Indicator $n$ refers to the sampling date, $m$ refers to different solutes $X$ (= $Ca^{2+}$, $Mg^{2+}$, $Na^+$, $K^+$, $Cl^-$), and brackets refer to concentrations. $W$ is the $n$ x $p$ matrix whose cell entries are proportions, $\alpha$, for each endmember in each stream sample. Again, $n$ refers to sampling dates, but $p$ is the number of sources of solutes (referred to as endmembers). The proportions refer to the fractions of sulfate in each sample that derive from an individual endmember, where the sum of proportions must equal $1 \pm 0.05$ for each sample. To derive the mixing proportions of sulfate specifically, we set up the NMF approach by normalizing each analyte concentration by sulfate concentration ($Y = SO_4^{2-}$), the target solute. After running the

algorithm for each of the three watersheds, we then inferred by inspection (see discussion below) that the endmembers represent different flowpaths in the subsurface. Therefore, these proportions of sulfate are referred to here as shallow, moderately shallow, and deep flowpaths, i.e. $\alpha_{shallow}$, $\alpha_{moderate}$, and $\alpha_{deep}$ respectively (only one of our target watersheds revealed the moderate-depth flowpath). $H$ is the $p$ x $m$ matrix whose cell entries are the concentration ratios that define the chemical signature of each of the $p$ endmembers. The key to NMF is that these concentration ratios are not determined prior to apportionment but rather are determined from the data itself. In addition, the chemical signatures of each endmember can vary temporally around central tendencies. Because the solution to eq. 1 is non-unique, we run the model 20,000 times, apply a filter to the models, and then calculate the mean and standard deviation of the remaining models for trend and error analysis (see SM section 1; Eq. S1).

The only hyperparameter that must be defined to run NMF a priori is the number of endmembers, $p$. We used principal component analysis (PCA) to determine the minimum number of components needed to explain >90% of the variance in stream solute ratios, and trained NMF to the bootstrapped data while assuming that number of endmembers. Machine learning determined the compositions defining the endmembers and the mixing proportions of each endmember in each sample. After running NMF, we interpreted each endmember composition based on geological and watershed knowledge.

Based on the outputs of the NMF model, we calculated the weathering rates of sulfide, carbonate, and silicate minerals in the watersheds. Additionally, we calculated the relative contributions of sulfuric and carbonic acid driving those weathering reactions. For details on the weathering calculations see SM section 2.

**2.4 Synthetic Dataset**

NMF is an algorithm that has been used for many applications (e.g., spectral analysis, email surveillance, cluster analysis; Berry et al., 2007) but has only recently been applied to stream chemistry (e.g., Xu and Harman, 2020). To exemplify the validity of our modeling approach, we generated a dataset of synthetic stream chemistry versus time and ran it through our NMF model. First, we defined two known endmember compositions, which are shown in Table S1. Next, we randomly generated 300 synthetic stream samples that were each calculated as a mixture of the two endmembers. Lastly, we ran NMF on the synthetic stream chemistry to determine the mixing proportions ($\alpha$) and endmember compositions ($[X]/[SO_4^{2-}]$), for all $X$.

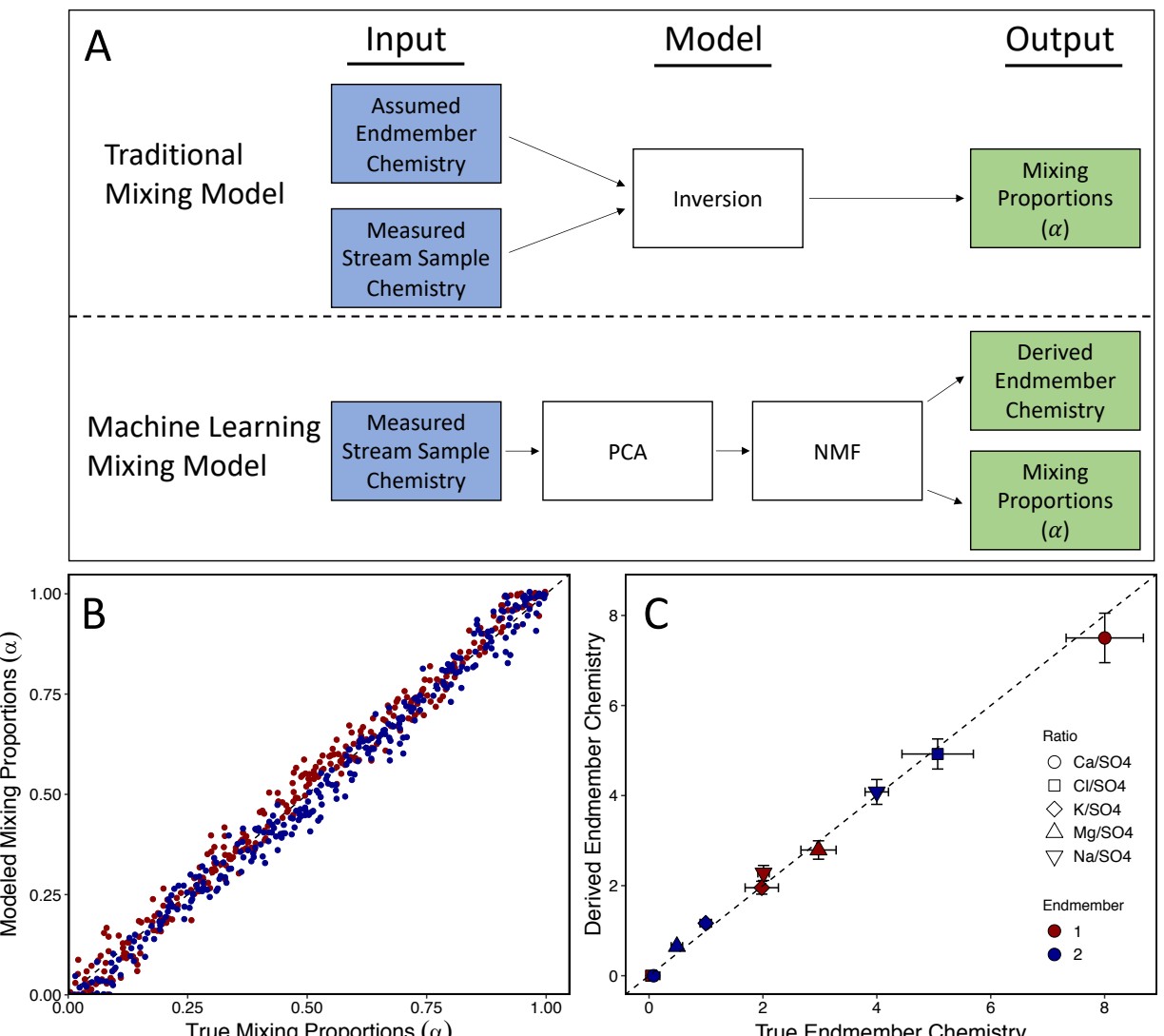

**Figure 2. Schematic diagram showing the differences between a traditional mixing model and our machine learning mixing model (A). Notably, in the machine learning mixing model, endmember chemistry is not assigned *a priori*, but rather derived from patterns in the data. Results from using our machine learning mixing model (i.e., NMF) on a synthetic dataset of known endmember chemistry and mixing proportions (i.e., α) are shown in B and C. Using only the synthesized stream sample chemistries, the model adequately recovered the correct mixing proportions (B) and endmember chemistries (C). The axes in (C) are the true concentration ratios of the endmembers and the NMF-derived concentration ratios of the endmembers.**

## 3 Results and Discussion

### 3.1 Synthetic Data Model

After generating the synthetic dataset of stream samples, we utilized NMF to determine the mixing proportions and endmember compositions. We then filtered out the poor fitting models (see SM eq. S1). As described more fully in the SM, this left an average number of valid models of 62 (range: 42-77). The average variance between valid models was <10%. Without any prior information about the system, NMF accurately determined the correct mixing proportions (RMSE = 0.04; $R^2$ = 0.98; $p$ < 0.001; Fig. 2B) and endmember compositions (RMSE = 0.21; $R^2$ = 0.99; $p$

< 0.001; Fig. 2C). In effect, the model was able to use patterns in the data to deconvolve sample chemistry into endmembers and proportions.

### 3.2 Application to Shale Hills

While clay minerals in shale-underlain watersheds in rainy climates are found at all depths because of their low chemical reactivity, pyrite and carbonate minerals are often chemically removed from upper layers and only found in unweathered shale at depth (Fig. 3; Brantley et al., 2013a; Wan et al., 2019; Gu et al., 2020a). For example, at Shale Hills, pyrite and carbonate minerals are only observed deeper than at least 15 meters below land surface (mbls) under the ridges and 2 mbls under the valley. In these deeper zones, calcite ($CaCO_3$), ankerite ($Ca(Fe_{0.34}Mg_{0.62}Mn_{0.04})(CO_3)_2$), and pyrite ($FeS_2$) dissolve in regional groundwaters that flow to the stream (Brantley et al., 2013a; Gu et al., 2020a). These groundwaters thus contribute DIC, $Ca^{2+}$, $Mg^{2+}$, and $SO_4^{2-}$ into the stream.

Like many catchments, water also flows to the stream in Shale Hills along a much shallower near-surface flowpath, which we call interflow (Fig. 3). Interflow is thought to occur along a transiently perched water table that lies within the upper 5-8 mbls. The most abundant mineral, illite ($K_{0.69}(Si_{3.24}Al_{0.76})(Al_{1.69}Fe^{3+}_{0.10}Fe^{2+}_{0.16}Mg_{0.19})O_{10}(OH)_2$), dissolves in interflow where it flows through the soil, with minimal illite dissolution in underlying weathered rock. Illite dissolution releases DIC and $Mg^{2+}$ and $K^+$ to interflow waters and causes precipitation of clays and iron oxides. Interflow derives ultimately from local precipitation that also contains $Na^+$, $Cl^-$, and $SO_4^{2-}$. Interflow and deep groundwater flowlines converge under the catchment outlet where the stream, on average, is 90% interflow and 10% deep groundwater (Sullivan et al., 2016; Li et al., 2017).

Only one mineral, chlorite ($(Fe^{2+}_{0.40}Mg_{0.15}Al_{0.35})_6(Si0_{.76}Al_{0.24})_4O_{10}(OH)_8$), is observed to begin to weather in the deep groundwater and continue weathering all the way to the surface (Fig. 3; Gu et al., 2020a). Chlorite thus dissolves to release $Mg^{2+}$ to both interflow and deep groundwater. While most water entering the catchment leaves as interflow without entering deep groundwater, the wide reaction zone observed for chlorite is consistent with a small fraction of water infiltrating vertically to the deeper zone (Brantley et al., 2017).

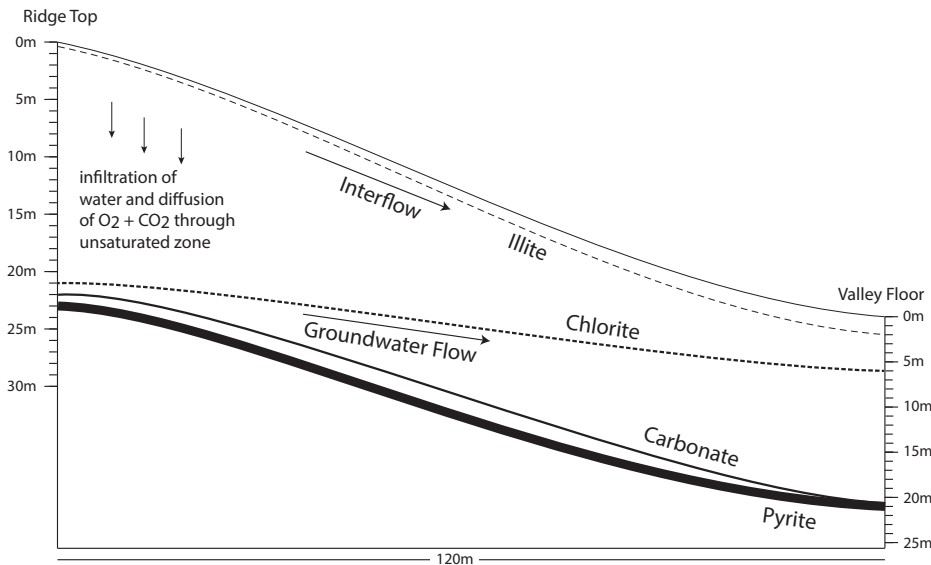

**Figure 3. Schematic cross section of Shale Hills showing the depths (labelled lines) where oxidation of pyrite, and dissolution of carbonate, chlorite, and illite initiate (modified after Brantley et al., 2013a). Illite and chlorite dissolve at all depths above the labelled lines, but reactions of carbonate minerals and pyrite only occur in a narrow one-meter wide depth zone under the ridge that widens to several meters toward the valley. Specifically, pyrite oxidation is complete under both ridge and valley at the depths where chlorite dissolution initiates. Carbonate dissolution is complete at the depth where pyrite oxidation is complete under the ridge but at ~4 m above the pyrite front under the valley. These reaction fronts are estimated and extrapolated from bulk chemistry measured in samples from boreholes located at the ridge and valley (Jin et al., 2010; Brantley et al., 2013a; Gu et al., 2020b).**

PCA for stream chemistry (2008-2010) at Shale Hills revealed two sources of sulfate, and this was used to set up NMF, i.e., $p = 2$ (Table S2). By comparing the compositions from matrix $H$ (Table S2) determined by NMF to our knowledge of the subsurface (Fig. 3), we interpreted the two endmembers as deep and shallow weathering along the two flowpaths, i.e. groundwater and interflow (Fig. 3), respectively (Jin et al., 2014; Sullivan et al., 2016). The endmember with high $[Ca^{2+}]/[SO_4^{2-}]$ and $[Mg^{2+}]/[SO_4^{2-}]$ was attributed to deep weathering because Ca- and Mg-containing minerals (i.e., calcite and ankerite) only dissolve at depth (Fig. 3; Jin et al., 2014; Gu et al., 2020a). The high $[Cl^-]/[SO_4^{2-}]$ endmember was attributed to shallow interflow because it is dominated by Cl-containing acid rain. This attribution revealed, consistent with other studies of the acid rain-impacted northeastern United States, that precipitation accounts for the majority of sulfate flux (i.e., 77%) at Shale Hills between 2008 and 2010.

Many lines of evidence back up these endmember attributions. The sulfate in the shallow endmember derives from interflow well above the pyrite oxidation front through pyrite-depleted rock and is thus attributed to acid rain, while the sulfate in the deep endmember is attributed mostly to pyrite oxidation. Some sulfate from acid rain may infiltrate to the regional groundwater, but the fraction is small. At Shale Hills, acid rain always contains $Cl^-$ and pyrite oxidation always preferentially dissolves carbonate minerals, giving each flowpath endmember a unique signature.

To test the NMF deconvolution, we compared these attributions to isotopic data. The value of $\delta^{34}S$ in dissolved sulfate is observed to correlate with increasing concentrations of pyrite-derived sulfate determined by NMF (Fig. 4A), consistent with depleted $\delta^{34}S$ signatures in pyrite (e.g., -20‰; Killingsworth et al., 2018). In contrast, acid rain shows $\delta^{34}S$ values around +3-5‰ (Bailey et al., 2004), and low sulfate concentrations in stream samples are characterized by $\delta^{34}S$ values within this range. Also, as pyrite oxidizes, the concentration of sulfate increases and the $\delta^{34}S$ values decrease to reflect the inferred composition of pyrite, -9.5‰ to -7.2‰ (Fig. 4A). Finally, Gu et al. (2020b) showed that pyrite oxidation drives the carbonate dissolution at Shale Hills. NMF results show that stream water was near calcite equilibrium (i.e., $\Omega_{calcite} = 1$; $\log \Omega_{calcite} = 0$) and had the highest pyrite-derived sulfate concentrations when the stream was fed by groundwater (Fig. 4B).

However, the annual flux of acid rain-derived sulfate from 2008-2010 in the shallow endmember determined from NMF at Shale Hills (Table 1) far exceeds the wet deposition of sulfate during the sampling period (Fig. 4C). Such inconsistencies have been noted elsewhere and attributed to travel-time delays over decades between acid rain input and stream output (Cosby et al., 1985; Prechtel et al., 2001; Mörth et al., 2005; Rice et al., 2014). Fig. 4C thus

allows us to estimate a ~19-31-year lag time between input and export of sulfate from the temporally changing acid rain (see SM section 2.4).

Weathering profiles at Shale Hills, the chemistry of the composition (*H*) matrix, sulfur isotopes, calcite saturation, and lag in acid rain export all support our interpretation that the two components in the NMF model are shallow and deep flowpaths and that sulfate largely derives from acid rain and pyrite respectively. The dissolution of
different minerals along these flowpaths lead to patterns in stream chemistry that our NMF model discerns and separates. If mineral reaction fronts are not separated in the subsurface, different flowpaths might not be separated by NMF; however, Brantley et al. (2017) and Gu et al. (2020a) have shown that separation of reaction fronts is common.

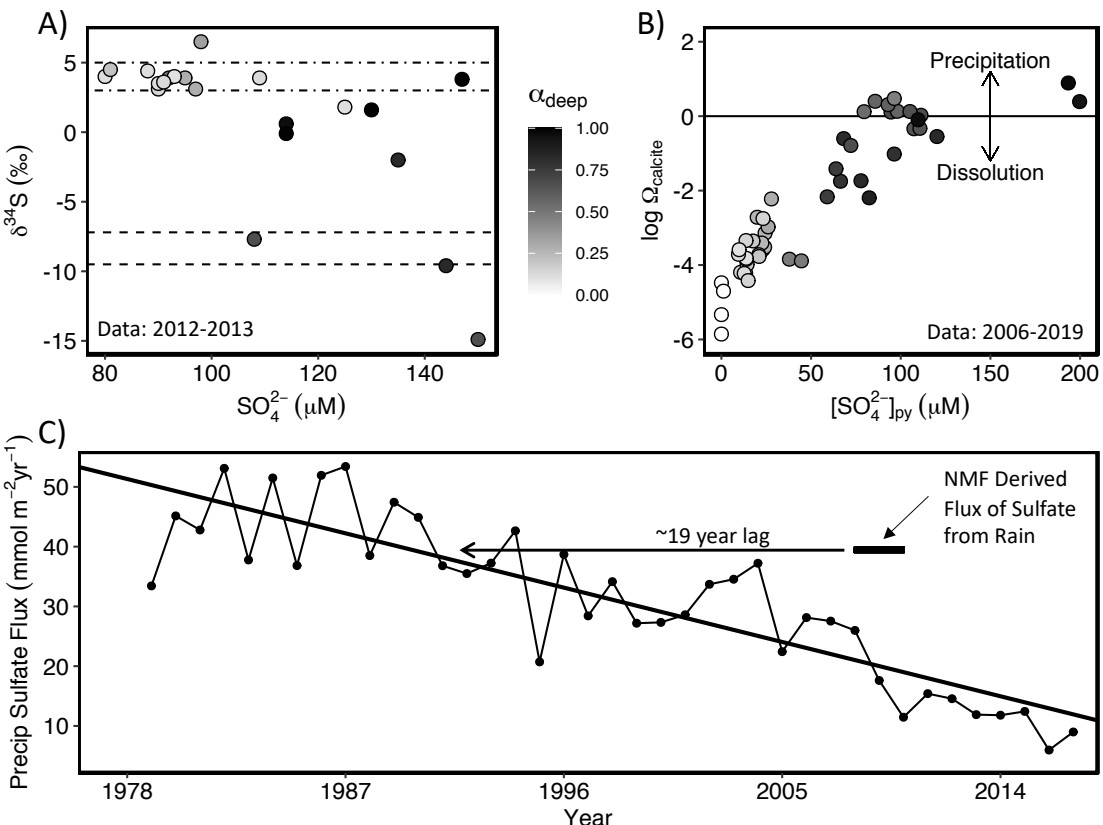

**Figure 4. (A) Sulfur isotope composition plotted versus concentration for sulfate in the subset of stream or groundwater samples at Shale Hills where S isotopes were measured (symbols; Jin et al., 2014). Dot-dashed lines represent the average sulfur isotope range for acid rain in USA (3-5‰; Bailey et al., 2004) and dashed lines represent the average sulfur isotope range of pyrite calculated from NMF results (-9.5‰ to -7.2‰). Sulfur isotopes in pyrite at Shale Hills were previously constrained to lie in the range of -1‰ to -15‰ (Jin et al., 2014). (B) Plot showing the calcite saturation index (log $\Omega_{calcite}$) vs. concentration of pyrite-derived sulfate (calculated through**
**NMF) in surface and groundwater samples at Shale Hills where alkalinity was measured. Here $\Omega_{calcite}$ (= ion activity product / equilibrium constant for calcite dissolution) is <1 the water is undersaturated with respect to calcite, and when $\Omega_{calcite}$ is >1, the water is oversaturated. Black line represents water-calcite equilibrium. Some samples in B differ from those in A because more samples were collected for alkalinity than sulfur isotopes. In both A and B, color shading represents the fraction of total sulfate derived from pyrite calculated by NMF (i.e., $\alpha_{deep}$). (C) Time series plot showing the flux of sulfate in Pennsylvania NADP site PA42 (2.8 km from Shale**
**Hills) from wet and dry deposition (see SM section 2.4). Black bar shows the NMF results for the export flux of sulfate derived from acid rain for Shale Hills during our sampling period, and the rationale for the inferred 19 y lag between input and output.**

### 3.3 Rates of Weathering and $CO_2$ Sequestration at Shale Hills

With these calculations we can use NMF results to elucidate the effect of sequestration or release of $CO_2$ at Shale Hills. We emphasize fluxes of importance over $10^5$– $10^6$ yr timescales. $CO_2$-driven weathering of the silicate minerals chlorite and illite removes carbon from the atmosphere and carries it as DIC in rivers to the ocean where it is buried as carbonate minerals (akin to reaction 2 in Figure 1, Table S3). In contrast, calcite and ankerite weathering coupled to pyrite oxidation instead releases $CO_2$ to the atmosphere over those timescales (reaction 7 in Figure 1) and

carbonate mineral weathering is neutral over those timescales (reaction 4 in Figure 1). Additionally, acid rain can interact with silicate minerals but not carbonate minerals at Shale Hills (because these are not present in the shallow subsurface (Fig. 3)). Thus $H_2SO_4$-dissolution caused by acid rain competes with $CO_2$-dissolution for silicates. This competition lowers the $CO_2$ consumption from silicate weathering, which been observed in other watersheds (e.g., Suchet et al., 1995).

To summarize the effect of weathering on $CO_2$ considered at the timescale of $10^5$– $10^6$ yr as shown in Figure 1, we propose a new parameter, the stream $CO_2$ sequestration coefficient, $\kappa_{stream}$ (see SM section 2.2 for full derivation). This coefficient is defined as mol $CO_2/[\Sigma^+]_{total}$ where $[\Sigma^+]_{total}$ is the sum of the equivalents of base cations in a sample. Here, equivalents refer to molar concentration multiplied by charge for an ion. Positive $\kappa_{stream}$ implies the stream acts as a source and negative implies it acts as net sink of $CO_2$ and the values are calculated for an

individual sample or integrated over some time period of stream sampling. The product of $\kappa_{stream}$ times $[\Sigma^+]_{total}$ in a sample equals the moles of $CO_2$ sequestered or released during weathering as represented in that sample (but the accounting is calculated for the reactions considered for the $10^5$-$10^6$ y timescale in Figure 1). Quantitatively this parameter reveals the moles of $CO_2$ sequestered or released during weathering per cation equivalent in a given stream sample:

$$\kappa_{stream} = \frac{1}{2}(-1 + \gamma_{stream} + \zeta_{stream}), \tag{2}$$

Here, $\gamma_{stream}$ is the proportion of cation equivalents in the stream derived from carbonate weathering per $[\Sigma^+]_{total}$, and $\zeta_{stream}$ is the ratio of sulfate equivalents from sulfuric acid per total base cation equivalents. We calculate $\gamma_{stream}$ for a sample by multiplying the pyrite-derived sulfate concentration (i.e., $\alpha_{deep}$ multiplied by total sulfate concentration) by the $[Ca^{2+}]/[SO_4^{2-}]$ and $[Mg^{2+}]/[SO_4^{2-}]$ ratios in the sample calculated by NMF to have derived from the deep

weathering endmember and then dividing by $[\Sigma^+]_{total}$. Likewise, $\zeta_{stream}$ is calculated by multiplying the fraction of sulfate from pyrite + acid rain (e.g., $\alpha_{deep} + \alpha_{shallow}$) by the total sulfate concentration and dividing that by $[\Sigma^+]_{total}$. This calculation shows that seasonally, Shale Hills switches between net source and net sink of $CO_2$ (Fig. 5D). Using the weathering reactions described in SM section 2.2, we also calculated the actual associated $CO_2$ fluxes; annual $CO_2$ dynamics are net-neutral at Shale Hills when considered over timescales of $10^5$– $10^6$ yr (Table 1; Fig. S4).

The switch in systems from operating as a source or a sink is attributed to seasonality in the dominant flowpath: $CO_2$-weathering of silicates occurs year-round, but $H_2SO_4$-weathering is more important in the wet season and is dominated by acid from rain. Specifically, in the dry season when water tables are low, the stream water is

often dominated by deeper groundwater flow that interacts with the deep pyrite reaction front and has little contribution of acid rain. However, even though this dry season is characterized by higher proportions of pyrite-derived sulfate, the watershed acts predominantly as a sink of $CO_2$ during this time of the year because the drawdown of $CO_2$ from $CO_2$-weathering of silicates is larger than the efflux of $CO_2$ from pyrite-driven $H_2SO_4$-weathering of carbonate (Fig. 5D). In the wet season when water tables are high, however, the stream is dominated by shallow interflow that does not interact with pyrite but has a large contribution of $H_2SO_4$ from rain. Kanzaki et al. (2020) also previously showed that the separation of reaction fronts (Fig. 3) can cause such important effects on $CO_2$ fluxes, although that previous treatment focused strictly on simple model systems unaffected by acid rain.

To test the accuracy of these inferences based on NMF, we compare to previous results for Shale Hills. Based on soil pore-water chemistry and rain fluxes at Shale Hills, Jin et al. (2014) estimated the $CO_2$ drawdown from silicate weathering to be 44 mmol m$^{-2}$ yr$^{-1}$. We find that if we assume all silicate weathering is $CO_2$-driven, then the silicate weathering drawdown is 38 mmol m$^{-2}$ yr$^{-1}$, which is consistent with the estimate of Jin et al (2014). But 44 mmol m$^{-2}$ yr$^{-1}$ is an overestimate because it does not consider $H_2SO_4$-weathering of silicates or carbonates.

**Table 1. Fluxes of $SO_4^{2-}$, Cations, and $CO_2$**

| | Shale Hills | East River | Hubbard Brook |
|---|---|---|---|
| | *Base Cation Fluxes (meq m$^{-2}$ yr$^{-1}$)*[a] | | |
| Total base cation flux | 336 ± 13 | 1540 ± 30 | 84.6 ± 0.8 |
| Base cation flux from $CO_2$-weathering of silicates | 12.6 ± 21.1 | 315 ± 58 | 24.1 ± 0.8 |
| Base cation flux from $CO_2$-weathering of carbonates | 216 ± 16 | 587 ± 48 | NA[c] |
| Base cation flux from $H_2SO_4$-weathering of silicates | 62.4 ± 1.0 | 152 ± 4 | 60.5 ± 0.2 |
| Base cation flux from $H_2SO_4$-weathering of carbonates | 44.8 ± 1.9 | 488 ± 9 | NA[c] |
| | *Fluxes (mmol m$^{-2}$ yr$^{-1}$)*[b] | | |
| Total sulfate flux | 50.3 ± 0.3 | 198 ± 1 | 30.3 ± 0.1 |
| Sulfide-derived sulfate flux | 11.2 ± 0.9 | 122 ± 4 | 9.1 ± 0.1 |
| Rain-derived sulfate flux | 38.9 ± 1.0 | 76.0 ± 4.2 | 21.2 ± 0.6 |
| $CO_2$ sequestration or release | 4.9 ± 10.7 | -35.6 ± 30.4 | -12.1 ± 0.4 |
| | *$CO_2$ Sequestration Coefficients* | | |
| $K_{stream}$[d,e] | 0.01 ± 0.03 | -0.02 ± 0.02 | -0.14 ± 0.01 |
| $K_{rock}$ | -0.08 ± 0.11 | 0.08 ± 0.17 | -0.19 ± 0.11 |

[a]Weathering fluxes calculated following procedure in SM section 2.2
[b]Negative $CO_2$ flux indicates sequestration and positive indicates release to atmosphere as considered over $10^5$– $10^6$ yr timescales (see Fig. 1)
[c]No carbonate cation fluxes reported because the bedrock contains no carbonate
[d]Stream $CO_2$ sequestration coefficient integrated over the period of record for each site
[e]Rock and stream $CO_2$ sequestration coefficients show that Shale Hills and East River are within error of net-neutral with respect to $CO_2$ and Hubbard Brook sequesters $CO_2$.

**3.4 East River**

Shale Hills is unique in that it is a monolithologic catchment and the data volume to constrain endmember apportionment is large. But NMF also works well for watersheds in which the subsurface flow and reactions are less constrained partly due to the more complex subsurface geology. The weathering profile at East River (underlain by

black shale) shows that pyrite and carbonate are depleted in upper layers but start dissolving at ~2-4mbls (Wan et al., 2019). PCA shows that the number of components is 2. The composition of the endmembers for East River are similar to Shale Hills (Table S2); however, the endmember composition indicates a higher proportion of $H_2SO_4$-weathering of carbonates (see SM section 2).

Based on NMF for East River, pyrite contributes 62% of the annual sulfate flux (Table 1). Sulfuric acid drives 29% to 69% of carbonate dissolution depending on the season, and this compares well with previous estimates of 35-75% (Winnick et al., 2017). Unlike Shale Hills, pyrite oxidation at East River is the dominant source of sulfate because acid rain is less important, and the black shale is pyrite-rich (Fig. 5B).

Although East River is like Shale Hills in that it intermittently switches between acting as a source or sink of $CO_2$ (Fig. 5), the seasonality of the switch between Shale Hills and East River is reversed. During baseflow (i.e., between periods of precipitation), Shale Hills is predominantly a sink of $CO_2$, and it sometimes switches to a source of $CO_2$ in the wet season because acid rain competes with $CO_2$ and reduces $CO_2$ consumption from silicate weathering. Without the large acid rain influx, East River instead acts as a sink of $CO_2$ during the wet season of snowmelt and then switches to a source during baseflow. Our results are consistent with previous interpretations (Winnick et al., 2017) suggesting $CO_2$ efflux rates are highest in baseflow-dominated and lowest in snowmelt-dominated flow regimes.

## 3.5. Hubbard Brook

Monolithologic shale watersheds are not the only target chemistries that can be deconvoluted with NMF: we now consider Hubbard Brook, a catchment on crystalline rock. Large variations in the $\delta^{34}S$ composition of the bedrock at Hubbard Brook (Bailey et al., 2004) mean that sulfur isotopes in stream water cannot be used to unambiguously apportion sulfate sources. Weathering fluxes from sulfide minerals are therefore difficult to constrained (Mitchell et al., 2001).

At Hubbard Brook, PCA shows three endmember sources of sulfate. As described below, we attribute these to three inferred flow lines, two in till and one at depth: waters flowing through i) shallow soil developed from till, ii) moderately-deep, less-weathered till, and iii) weathering bedrock. A three-layered weathering profile has been observed in other till-covered areas of New Hampshire as well (Goldthwait and Kruger, 1938). We used these ideas to identify endmembers as described below.

Concentrations of sulfate in acid rain have declined over time in northeastern USA (Lynch et al., 2000; Lehmann et al., 2007). Of the three NMF-determined endmembers at Hubbard Brook, two of them show declining sulfate concentrations with time. We therefore attributed the first and second endmembers to acid rain (Fig. S1).

Only one endmember showed little to no decline in sulfate concentration over time, and we therefore attributed that endmember to deep weathering in water interacting with the underlying bedrock. The composition of the deep weathering endmember shows a strong correlation between $[Mg^{2+}]/[SO_4^{2-}]$ and $[K^+]/[SO_4^{2-}]$. This chemical signature is similar to previous observations of weathering of metasedimentary rock piles where silicates (biotite and chlorite) are the first minerals to dissolve when sulfides oxidize (Moncur et al., 2009). Specifically, biotite $(K(Si_3Al)Mg_2FeO_{10}(OH)_2)$ is known to release $Mg^{2+}$ and $K^+$ while chlorite releases $Mg^{2+}$ upon weathering.

Moreover, the metamorphic conditions that produce pyrrhotite also produce biotite and chlorite, and those three minerals tend to be located together in schist foliations (Carpenter, 1974). We thus infer that pyrrhotite oxidation at Hubbard Brook apparently causes dissolution of biotite ± chlorite because these are the most susceptible minerals in close proximity to the sulfide. Thus, several lines of evidence underlie our interpretation that component 3 is the deep weathering source of sulfate.

From the NMF results summarized in Table 1, pyrrhotite can account for 30% of the total sulfate flux at Hubbard Brook. The schist and till contain essentially no carbonate; therefore, weathering is always a net sink for $CO_2$. In this watershed, however, the story is complicated by the dissolution of silicate minerals by sulfuric acid from pyrrhotite oxidation and acid rain. If we had assumed all of the base cations detected in Hubbard Brook were caused by $CO_2$-weathering, we would have overestimated the net drawdown of $CO_2$ out of the atmosphere (Fig. 1).

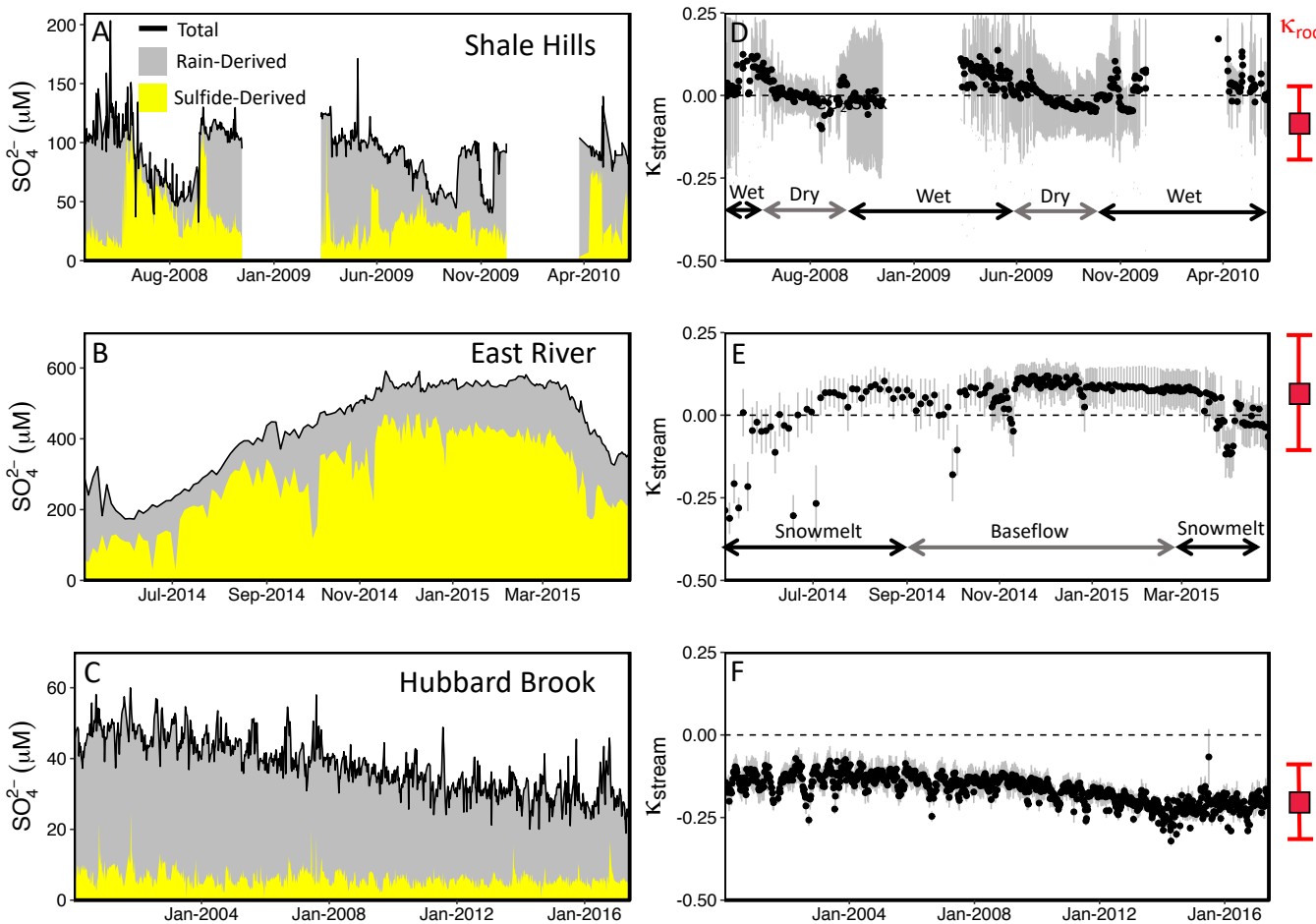

**Figure 5. Concentration of total sulfate (black line), rain-derived sulfate (NMF-calculated; gray) and sulfide-derived sulfate (NMF-calculated; yellow) in stream water plotted versus time at Shale Hills (A), East River (B), and Hubbard Brook (W-3 sub-catchment) (C). Shale Hills and East River temporally switch between being a source and sink of $CO_2$, while Hubbard Brook is always a sink over the timescales studied, as shown by the $CO_2$ sequestration coefficient ($\kappa_{stream}$) for Shale Hills (D), East River, (E), and Hubbard Brook (F). Gray error bars in D, E and F represent 1s.d. from the calculated $\kappa_{stream}$ for that sample. The range (mean + 1s.d.) indicated in red to the right of D, E, and F represent $\kappa_{rock}$, the time integrated $CO_2$ sequestration coefficient calculated from the rock chemistry (see text). Here, $\kappa_{stream} > 0$ or $<0$ indicates stream is a source or sink of $CO_2$ respectively when considering weathering reactions over $10^5$ to**

 **$10^6$ yr timescales (see Fig. 1). The long record at Hubbard Brook shows that $\kappa_{stream}$ is approaching $\kappa_{rock}$ as the watershed recovers from acid rain. Gaps in the time series for Shale Hills occur when the autosampler tubing or stream froze.**

### 3.6 Predicting CO₂ release or drawdown from rock chemistry

From the stream chemistry, we found that Shale Hills and East River are net neutral with respect to $CO_2$, and
Hubbard Brook is a net sink (Table 1; Figure 5). In Table 1, the weathering fluxes are summarized as $CO_2$ fluxes (see
SM section 2.2; Fig. S4), but the NMF results can also be used to calculate weathering losses for each mineral as
described in SM 2.5 (Table S5). Although we do not explicitly discuss each of these mineral-related fluxes learned
from NMF, they have resulted in differences in composition of soil versus protolith and we can use soil chemistry
therefore as an additional test of $\kappa_{stream}$: specifically, we compare $\kappa_{stream}$ to the $CO_2$ flux recorded in the weathered
profile as solid-phase chemistry. To do this, we calculate a $CO_2$ sequestration coefficient analogous to $\kappa_{stream}$ but
instead based on rock chemistry, $\kappa_{rock}$, by assessing soil and taking into account the fraction of base cations
weathered, the fraction of base cations from carbonates, and the capacity of the bedrock to produce $H_2SO_4$:

$$\kappa_{rock} = \frac{1}{2}(\tau + \gamma_{rock} + \zeta_{rock}), \tag{3}$$

In effect, $\kappa_{rock}$ is the time-integrated $CO_2$ sequestration coefficient recorded as the solid phase weathering products in
units of mol $CO_2$/eq base cation. In eq. 3, $\tau$ is the mass transfer coefficient for base cations at the land surface (where
$1-\tau$ equals the fraction of total base cations originally present in parent rock that remain in topsoil at land surface),
$\gamma_{rock}$ is the proportion of base cations in the bedrock associated with carbonate minerals, and $\zeta_{rock}$ is the acid
generation capacity of the rock. The derivation of eq. 3 and description of each variable is more fully summarized in
SM section 2.3. Briefly, $\gamma_{rock}$ expresses the proportion of base cations in the parent rock that are associated with
carbonate minerals (varies from 0 to 1 for 100% silicate protolith to 100% carbonate protolith). $\zeta_{rock}$ expresses the
relative amount of (acid-generating) pyrite to base cations in the protolith (varies from 0 to 1.5 for catchments where
100% of weathering is $CO_2$-driven to catchments where 100% of weathering is $H_2SO_4$-driven, respectively). $\tau$
expresses the fraction of cations that have not dissolved away upon exposure at the land surface (varies from -1 to 0
for 0% cations remaining at land surface to 100% cations remaining, respectively). Negative $\kappa_{rock}$ describes a
lithology that has been net sequestering $CO_2$ over the duration of weathering, whereas positive $\kappa_{rock}$ has been net
releasing $CO_2$. Based on the chemistry of the bedrock and topsoil at each watershed, $\kappa_{rock}$ is $-0.08 \pm 0.11$, $0.08 \pm 0.17$,
and $-0.19 \pm 0.11$ for Shale Hills, East River, and Hubbard Brook, respectively (Tables 1, S4). Based on these values
from observations of the solid weathering phases, Shale Hills and East River on net are $CO_2$ neutral (i.e., within error
of 0), but Hubbard Brook has acted as a long-term $CO_2$ sink.

If the streams at each site today are acting just like the weathering recorded over the last tens of thousands of
years in the solid-phase material and our assumptions about $CO_2$- versus $H_2SO_4$-weathering are correct, $\kappa_{rock}$ should
equal $\kappa_{stream}$. Here, we find that $\kappa_{stream}$ (discharge-weighted average) for Shale Hills, East River, and Hubbard Brook
are $0.01 \pm 0.03$, $-0.02 \pm 0.02$, and $-0.14 \pm 0.01$ respectively (Table 1, Fig. 5). For all sites, the stream chemistry shows
similar values of $CO_2$ sequestration coefficient for the modern (stream timescale) compared to the time-integrated

(soil timescale), i.e., $\kappa_{stream} \approx \kappa_{rock}$, consistent with Shale Hills and East River acting as $CO_2$ net neutral but Hubbard

Brook as a $CO_2$ sink.  In addition, at Hubbard Brook, it can be seen that acid rain has competed with $CO_2$ in

weathering minerals, lowering the capacity of the rock to sequester atmospheric $CO_2$. Because our calculation of $\kappa_{rock}$

does not include acid rain, we would expect acid rain would increase $\kappa_{stream}$ relative to $\kappa_{rock}$, which is what we observe

at Hubbard Brook. Hubbard Brook has only moved back to equivalency between the rock and stream record in recent

years (2013-2016; Fig. 5F) as the system has recovered from acid rain. These comparisons also suggest that rock

chemistry, which is much easier to analyze, can sometimes predict stream fluxes adequately.

## 4 Conclusions

By not requiring a priori assignments of endmembers, our machine learning model not only successfully reproduced

source apportionments made in more traditional endmember analysis for streams, but also revealed new information

about how watersheds work.  At the same time, the method also solved some issues related to source apportionment

for streams with time variations of large acid rain inputs. The approach documented that two carbonate-containing

shale watersheds (Shale Hills, East River) are intermittent sources or sinks of $CO_2$ to the atmosphere but on net are

neutral with respect to $CO_2$. In contrast, because it has no carbonate minerals, Hubbard Brook is a constant sink for

$CO_2$ (Figs. 5 and S5). These observations were compared and confirmed by comparing stream chemistry to rock

chemistry.

NMF also emphasized the importance of different water flowpaths in determining endmembers: the

endmembers were not strictly defined by mineralogy but by patterns of subsurface flow that can be related to

subsurface reaction zones. These flowpaths lead to patterns in stream water chemistry that were easily deciphered by

our newly developed machine learning-based mixing model. In particular, for three streams, signals in the chemical

variations were observed to reveal dissolution of the most reactive mineral in proximity to sulfide oxidation. Many

watersheds have flowpaths distinguished by geochemical signatures from mineral reactions (Brantley et al., 2017)

but we do not know these paths a priori when we investigate stream chemistry. Machine learning will be useful to

model mineral reactions on broader spatial scales and will help constrain global weathering-related $CO_2$ dynamics

because it can delineate endmembers without a priori assumptions.

Beyond these attributes, the machine learning approach also revealed other new attributes of weathering. In

Shale Hills, we discovered that sulfate inputs from acid rain may not be exported completely for two decades, which

impacts mass balance and weathering-related $CO_2$ dynamics. Although not discussed explicitly here, this decadal

time-lag was also observed at Hubbard Brook. NMF also showed that Hubbard Brook, recovering from the impacts

of acid rain, is only recently returning to its full potential as a $CO_2$-sequestering rock system. In other words, prior to

acid rain, Hubbard Brook sequestered more $CO_2$ per mole of weathered bedrock than it does today. But acid rain

dissolved some of the silicates with $H_2SO_4$, lowering the $CO_2$ sequestration capability of the watershed. NMF led us

to discover this new attribute of acid rain, namely that it diminishes the capacity of a rock to sequester $CO_2$ at

millennial timescales (Figure 1) by replacement of $CO_2$ by $H_2SO_4$ as a weathering agent. Regardless of the net $CO_2$

dynamic, we discovered that without considering sulfide oxidation or acid rain, the $CO_2$ weathering sink considered over $10^5$ to $10^6$ yr timescales is always overestimated.

**Acknowledgements**

Financial support for this work was predominantly provided by National Science Foundation Grant EAR 13-31726 to
SLB for the Susquehanna Shale Hills Critical Zone Observatory. The Shale Hills catchment is located in Penn State's Stone Valley Forest, which is funded by the Penn State College of Agriculture Sciences, Department of Ecosystem Science and Management, and managed by the staff of the Forestlands Management Office. XG acknowledges support from Office of Basic Sciences Department of Energy grant DE-FG02-05ER15675 to SLB; TW acknowledges support from Penn State College of Earth and Mineral Sciences Distinguished Postdoctoral Scholar
Fellowship and NSF grant IIS 16-39150 to SLB and Zhenhui Li; ARS acknowledges support from a Penn State University Graduate Fellowship and NSF Graduate Research Fellowship.

**Data Availability**

Data used in analysis for this work can be found at the Susquehanna Shale Hills Critical Zone website

(http://www.czo.psu.edu/data_time_series.html), the Hubbard Brook data catalog

([https://hubbardbrook.org/d/hubbard-brook-data-catalog](https://hubbardbrook.org/d/hubbard-brook-data-catalog)), and in Wan et al. (2019) and in Winnick et al. (2017).

Codes used in the modeling are available upon request.


**Author Contributions**

SLB, TW, and ARS conceived the project. TW and ARS developed the codes for the model. SLB, XG, and ARS interpreted the data and model outputs. All authors contributed in the preparation of the manuscript.

**Competing Interests**

The authors declare that they have no conflict of interest.

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
