# Peer review of "Machine Learning Deciphers CO2 Sequestration and Subsurface Flowpaths from Stream Chemistry"

_Hydrology and Earth System Sciences, 2020_

## Referee Comment (RC1) · Anonymous Referee #1 · 25 Nov 2020

General comments

The paper applies NMF (Non-negative Matrix Factorization), which is a machine learning technique, to EMMA (End-Member Mixing Analysis). They use this to calculate CO2 sequestration in three watersheds. The novelty is the application of NMF to EMMA. In general, the paper is well written. I suggest publication if the comments below are addressed.

Specific comments

1. Line 19-20, 44-45 and 412. You talk about a "new machine learning technique". Actually, it is not a new technique. What you do is applying an old technique (machine

learning or, more specifically, NMF) to EMMA, which is new.

2. Line 132-134. You say that NMF is unique in that it does not rely on assumptions of endmembers a priori. This is repeated throughout the whole paper (figure 2, line 172, 412 and 428). I think this is not entirely true. For instance, Carrera et al. (2004) calculate endmembers without NMF. Carrera, J., E. Vázquez-Suñé, O. Castillo, and X. Sánchez-Vila (2004), A methodology to compute mixing ratios with uncertain endmembers, Water Resour. Res., 40, W12101, doi: 10.1029/2003WR002263.

3. Line 138: You use SO4 as a reference for solute concentrations. To me it would make more sense to use Cl-, instead, because it is not likely involved in chemical reactions. Is there a particular reason for using SO4?

4. Line 145: You define end members for shallow, moderately shallow, and deep flowpaths. Of course, they may vary in time as you say in line 149. Could this create some bias? For example, end members of deep flowpaths are generally older with water that fell as rainfall earlier than end members of shallow flowpaths. As acid rain varies with time, differences in chemical signature can be affected by the age of the water.

5. Line 265: Equation 3 and kstream are not clear to me. Where does the -1 come from? I suggest adding an explanation in the SM like you have done for krock.

6. SM, section 2.2. I find this section very hard to follow. Actually, you describe mathematical equations by using text. I think you can make it more readable, if you put the equations as well.

7. SM, line 55-57. If I understand correctly, here you attribute all Ca and Mg to carbonate dissolution. However, it can also come from silicates. In fact, in figure 1 you represent silicates by CaSiO2. Do you simply neglect Ca from silicate weathering?

Technical corrections

1. SM, line 28. Change "in in" to "in".

2. SM, line 133-134. I think this equation is equation 3 from the main text, not 2.

3. SM, line 148. You refer to Fig. 3C. However, this figure contains nothing related to lag times. I think you mean Fig. 4C.

---

## Referee Comment (RC2) · Anonymous Referee #2 · 23 Dec 2020

Review of "Machine Learning Deciphers CO2 Sequestration and Subsurface Flowpaths from Stream Chemistry" by Andrew R. Shaughnessy, Xin Gu, Tao Wen, and Susan L. Brantley

This study focuses on applying machine learning to endmember mixing analysis of weathering chemistry in subsurface groundwater flow paths. They apply an NMF scheme and train it on syntactic data generated using a multivariate normal distribution of log-transformed stream water chemistries. The NMF is then applied to 3 measured stream water samples to delineate mixing proportions. The study is well presented and written, the SM is seminal to the understanding of the study and holds the key details

for the optimization of the NMF. The main finding is within the sensitivity of the reaction to the groundwater flow paths which are unknown, yet they control the concentration relation between the components and therefore they are controlling the overall chemistry. In a way, these flow paths are a spatial localization of the reaction in space and time, due to the seasonal effects, as shown here. I found the paper very interesting, well written, and clear and supports the publication of the study, yet the main missing part that is not discussed here and must be added is a discussion on the how. How does the NMF manage to capture the effect of the subsurface groundwater flow paths? What is the additional mechanism that is deciphered by the NMF? The spatial and temporal effect of the subsurface groundwater flow paths must be captured in a mean-field way by the MNF, and this is not clear how it managed to do so and what was the missing mechanism.

I agree with referee 1 remarks 5 and 6, do clarify the mathematical components with a mathematical expression.

---

## Author Comment (AC1) · 20 Jan 2021

Response to Reviewer 1.

General comments:
The paper applies NMF (Non-negative Matrix Factorization), which is a machine learning technique, to EMMA (End-Member Mixing Analysis). They use this to calculate CO2 sequestration in three watersheds. The novelty is the application of NMF to EMMA. In general, the paper is well written. I suggest publication if the comments below are addressed.

We thank the reviewer for providing helpful comments towards improving the paper. We incorporated the comments suggested here, which we believe strengthened the manuscript. Below are the specific responses to each comment.

Specific comments

1. Line 19-20, 44-45 and 412. You talk about a "new machine learning technique". Actually, it is not a new technique. What you do is applying an old technique (machine learning or, more specifically, NMF) to EMMA, which is new.

We agree with the reviewer that the actual machine learning technique is not new; however, the application of NMF towards EMMA is new. We have revised the manuscript to highlight the novelty of the application rather than the technique, specifically in the lines mentioned above. For example, instead of "We apply a new machine learning technique in three watersheds to determine the minerals dissolved by each acid", we will re-phrase as "We use a new implementation of machine learning to EMMA in three watersheds to determine the minerals dissolved by each acid."

2. Line 132-134. You say that NMF is unique in that it does not rely on assumptions of endmembers a priori. This is repeated throughout the whole paper (figure 2, line 172, 412 and 428). I think this is not entirely true. For instance, Carrera et al. (2004) calculate endmembers without NMF. Carrera, J., E. Vázquez-Suñé, O. Castillo, and X. Sánchez-Vila (2004), A methodology to compute mixing ratios with uncertain endmembers, Water Resour. Res., 40, W12101, doi: 10.1029/2003WR002263.

We thank the reviewer for this reference. In general, since the inception of endmember mixing analysis (EMMA), many researchers have worked to improve the methodology. Hooper (2003) developed diagnostic tools for evaluating the fit of multivariate data to lower dimensional mixing space without explicitly identifying endmembers. Neal et al. (2010) developed an extended EMMA model that incorporated fluxes into the mixing calculations. Pelizardi et al. (2017) improved EMMA by incorporating chemical reactions and determining decoupled conservative constituents for the mixing calculations. Our understanding of the Carrera et al. (2004) method is that they utilize measured endmembers that are assumed to be noisy, under sampled, or incorrect, and they use their model to optimize the true endmember composition for the mixing ratio calculation. We will mention these prior contributions, including that of Carrera et al.

Our NMF model provides another improvement to EMMA, where patterns in stream chemistry alone are used to derive optimal endmember compositions. Because we make no

assumptions a priori, the model yielded endmembers that we inferred were consistent with mineral reactions along distinct flowpaths a posteriori. This is unlike other EMMA models. Based on the literature that we have read, we believe that there are no other EMMA approaches that start with no assumptions a priori, i.e., interpreting the identity of derived endmembers is a unique feature of NMF applied to EMMA. If such EMMA approaches are out there, we should cite them but we have not seen them.

We plan to add the following to the manuscript:
"EMMA has been used by hydrogeochemists to understand the sources of water and the transport of various solutes. Since its inception, many researchers have aimed to improve EMMA. One focus of improvements has been toward more accurately determining unknown or under-constrained endmember chemistry (Hooper, 2003; Carrera et al., 2004; Valder et al., 2012). But these efforts all use some a priori determination of endmembers. Our NMF model adds to the growing effort to improve EMMA by applying blind source separation."

References:

Hooper, R. P.: Diagnostic tools for mixing models of stream water chemistry, Water Resources Research, 39, 1055, 235 https://doi.org/10.1029/2002WR001528, 2003.

Neal, C., Jarvie, H. P., Williams, R., Love, A., Neal, M., Wickham, H., ... & Armstrong, L. (2010). Declines in phosphorus concentration in the upper River Thames (UK): Links to sewage effluent cleanup and extended end-member mixing analysis. *Science of the Total Environment*, *408*(6), 1315-1330.

Pelizardi, F., Bea, S. A., Carrera, J., & Vives, L. (2017). Identifying geochemical processes using End Member Mixing Analysis to decouple chemical components for mixing ratio calculations. *Journal of Hydrology*, *550*, 144-156.

Valder, J. F., Long, A. J., Davis, A. D., & Kenner, S. J. (2012). Multivariate statistical approach to estimate mixing proportions for unknown end members. *Journal of hydrology, 460*, 65-76.

3. Line 138: You use SO4 as a reference for solute concentrations. To me it would make more sense to use Cl-, instead, because it is not likely involved in chemical reactions. Is there a particular reason for using SO4?

We think there is a slight misconception here of our approach: we chose to use SO4 as the reference for the solute concentrations because we designed the NMF approach in this case to determine endmember sources of the SO4. Specifically, we were interested in solving for the mixing proportion ($\alpha$) of sulfate and so we set up the NMF approach to do that. In other words, we want to know how much sulfate came from each source. If we normalized our solute concentrations using Cl as a reference, then the mass balance works out such that we are solving for the mixing proportions of chloride. We could back-calculate the sulfate mixing proportions from the chloride mixing proportions following:

$$\alpha_{SO4,i} = \frac{\alpha_{Cl,i} \left(\frac{SO4}{Cl}\right)_i}{\left(\frac{SO4}{Cl}\right)_{riv}}$$

Here, $\alpha_{m,i}$ is the mixing proportion of element m and endmember i, and SO4/Cl is the concentration ratio of sulfate to chloride in endmember *i* or in the stream sample (i.e., subscript "riv"). This adds an additional complexity to the model by needing to switch between mixing ratio designations. Additionally, our analysis showed that the Cl/SO4 ratio of the deep weathering endmember/flowpath is 0 (Table S1). If we were to have normalized the dataset by Cl, we would have lost the deep weathering endmember because there is only one source of chloride in the watershed (i.e., rain).

We will include the following sentence in the main text of the manuscript for clarity. "To derive the mixing proportions of sulfate specifically, we set up the NMF approach by normalizing each analyte concentration by sulfate concentration, the target solute."

4. Line 145: You define end members for shallow, moderately shallow, and deep flowpaths. Of course, they may vary in time as you say in line 149. Could this create some bias? For example, end members of deep flowpaths are generally older with water that fell as rainfall earlier than end members of shallow flowpaths. As acid rain varies with time, differences in chemical signature can be affected by the age of the water.

We thank the reviewer for their thoughtful question. The time variance in acid rain chemistry is one reason why a traditional inverse approach fails to separate the sources of sulfate in our study sites and one of the motivating reasons for our approach using NMF. If we understand the comment of the reviewer correctly, they are concerned that different flowpaths inherently have different timescales and the endmember chemistry of acid rain also varies with time; thus, deeper flowpaths might only represent older acid rain rather than mineral weathering. This is a difficult question due to spatial and temporal heterogeneity in endmember compositions. We believe that our model captures the change in endmember chemistry because the endmember chemistry does have a time dependence. As seen in Fig. S3, the concentration ratios of the endmembers change over time. For example, Fig. S1 shows that sulfate concentrations associated with acid rain in Hubbard Brook decrease with time, which is consistent with measured rain chemistry; however, the chemistry of the sulfide mineral contribution is constant with time because it does not depend on the changing rain chemistry. When we revise our paper, we will endeavor to make this clear.

We believe that our apportioning of sulfate from acid rain and pyrite oxidation is accurate for several reasons. First, when our model shows that the stream is dominated by pyrite-derived sulfate, we observe sulfur isotopes that are consistent with pyrite oxidation rather than acid rain ($\delta^{34}S < 0$; Fig. 4A). Additionally, the co-occurrence of the pyrite and carbonate reaction fronts suggest that these minerals should weather along the same flowpath (Figs. 3 and 4B).

Although our evidence (outlined in section 3.2) is consistent with our interpretation that the model separates acid rain from pyrite oxidation, it is possible that some older inputs of acid rain are lumped into our pyrite fraction, i.e., if acid rain contributions infiltrate to deep groundwaters

where pyrite is found. This would make our pyrite-derived sulfate flux a maximum estimation of the weathering flux.

5. Line 265: Equation 3 and kstream are not clear to me. Where does the -1 come from? I suggest adding an explanation in the SM like you have done for krock.
6. SM, section 2.2. I find this section very hard to follow. Actually, you describe mathematical equations by using text. I think you can make it more readable, if you put the equations as well.

We thank the reviewer for these questions and we recognize we need to explain these concepts more clearly. The derivation of $\kappa_{stream}$ (comment 5) is better understood by first expanding section 2.2 to include explicit equations (comment 6). Because these two sections are connected, we will answer both comments 5 and 6 together.

Section S2.2 describes how we calculate the inferred $CO_2$ release or sequestration resulting from weathering as recorded in the sum of all cation concentrations (meq/l) in each stream sample, $[\Sigma^+]_{total}$:

$$[\Sigma^+]_{total} = 2[Ca^{2+}]_{total} + 2[Mg^{2+}]_{total} + [Na^+]_{total} + [K^+]_{total}, \qquad (S1)$$

To calculate the inferred $CO_2$ release or sequestration resulting from weathering, we use the results of NMF, as described below, to identify the extents of 4 weathering reactions recorded in each stream sample: 1) $CO_2$-driven weathering ($CO_2$-weathering) of silicates, 2) $H_2SO_4$-driven weathering ($H_2SO_4$-weathering) of silicates, 3) $CO_2$-weathering of carbonates, and 4) $H_2SO_4$-weathering of carbonates. We note these four quantities respectively as 1) $[\Sigma^+]_{carbonate-CO_2}$; 2) $[\Sigma^+]_{silicate-H_2SO_4}$; 3) $[\Sigma^+]_{silicate-CO_2}$; 4) $[\Sigma^+]_{carbonate-H_2SO_4}$. These are the four unknowns we seek to calculate for SH and ER, as described below.

Based on the high proton and low metal concentrations of the measured rain chemistry, the rain contributes negligibly to the base cation concentrations of the study streams; therefore, we apportioned all the base cations to weathering reactions. First, we note that the meq/l of cations derived from carbonate minerals, $[\Sigma^+]_{carbonate}$, equal $[\Sigma^+]_{carbonate-CO_2}$ + $[\Sigma^+]_{carbonate-H_2SO_4}$. Likewise, the meq/l of cations derived from silicate minerals, $[\Sigma^+]_{silicates}$ equal $[\Sigma^+]_{silicate-H_2SO_4}$ + $[\Sigma^+]_{silicate-CO_2}$. The summation of silicate-cations ($[\Sigma^+]_{silicate}$) is the difference between the summation of total cations ($[\Sigma^+]_{total}$) and that of carbonate-derived cations ($[\Sigma^+]_{carbonate}$):

$$[\Sigma^+]_{silicate} = [\Sigma^+]_{total} - [\Sigma^+]_{carbonate}, \qquad (S2)$$

We use a few field observations to complete the calculations for SH and ER. First, carbonate minerals only dissolve in water flowing along the deep path because carbonates have been depleted from shallow depths. Second, although some chlorite dissolves into water flowing along the deep path, the release of Mg is insignificant at depth compared to Mg released from carbonate. So we can ignore shallow dissolution of carbonates and deep dissolution of silicates in both SH and ER.

With these observations, we can write:

$$[\Sigma^+]_{carbonate} = 2\alpha_{deep}[SO_4^{2-}]_{total}\left(\left(\frac{[Ca^{2+}]}{[SO_4^{2-}]}\right)_{deep} + \left(\frac{[Mg^{2+}]}{[SO_4^{2-}]}\right)_{deep}\right), \tag{S3}$$

Here, $\alpha_{deep}$ is the proportion determined through NMF of sulfate in a given water sample that was derived from reactions along the deep flowpath, $[SO_4^{2-}]_{total}$ is the total concentration of sulfate in the stream water sample under consideration, $([Ca^{2+}]/[SO_4^{2-}])_{deep}$ and $([Mg^{2+}]/[SO_4^{2-}])_{deep}$ are the model-derived ratios of $[Ca^{2+}]$ and $[Mg^{2+}]$ to $[SO_4^{2-}]$, respectively, that characterize the deep flowpath endmember for that sample.

Remembering that Mg release from chlorite dissolution at depth is insignificant compared to Mg from carbonates, all of the generated sulfate in the deep weathering endmember is balanced by cations from dissolved carbonate minerals:

$$[\Sigma^+]_{carbonate-H_2SO_4} = 2\alpha_{deep}[SO_4^{2-}]_{total}, \tag{S4}$$

(We multiply the concentration of deep sulfate by 2 because 2 eq of cations are released per equivalent of sulfate.) Any carbonate-derived base cations that are in excess of what could have been produced by pyrite-derived sulfuric acid are attributed to $CO_2$-weathering of carbonates:

$$[\Sigma^+]_{carbonate-CO_2} = [\Sigma^+]_{carbonate} - [\Sigma^+]_{carbonate-H_2SO_4}, \tag{S5}$$

Remembering that no carbonates dissolve into water flowing along the shallow path, then similar arguments for the shallow flowpath yield:

$$[\Sigma^+]_{silicate-H_2SO_4} = 2\alpha_{shallow}[SO_4^{2-}]_{total}, \tag{S6}$$

$$[\Sigma^+]_{silicate-CO_2} = [\Sigma^+]_{silicate} - [\Sigma^+]_{silicate-H_2SO_4}, \tag{S7}$$

From these equations, values for the four unknowns can be calculated for SH and ER. A similar approach was taken for HB except that no carbonate minerals were present, and only two unknowns were determined ($[\Sigma^+]_{silicate-H_2SO_4}$, $[\Sigma^+]_{silicate-CO_2}$).

**Calculating $CO_2$ and $\kappa_{stream}$**
Considered over the long-term, $H_2SO_4$-weathering of silicates and $CO_2$-weathering of carbonates are $CO_2$ neutral, while $CO_2$-weathering of silicates sequesters $CO_2$ (1 eq $CO_2$ per 2 eq cations) and $H_2SO_4$-weathering of carbonates releases $CO_2$ (1 eq $CO_2$ per 4 eq cations; Fig. 1). For a given water sample, the cation concentrations record the extent of dissolution of carbonate and silicates, as long as the contribution of these base cations from acid rain is minimal. Therefore, the uptake or release of $CO_2$ recorded in a given sample, $\Delta CO_2$, can be calculated for any given stream water sample:

$$\Delta CO_2 = 0.5\,[\Sigma^+]_{silicate-CO_2} - 0.25\,[\Sigma^+]_{carbonate-H_2SO_4}, \tag{S8}$$

Now that we have expanded the supplemental section S2.2, we can more easily explain our derivation of $\kappa_{stream}$. What follows is a derivation of $\kappa_{stream}$.

In general, $\kappa_{stream}$ was defined to be comparable to $\kappa_{rock}$. Both are used as ways to note the capacity of a watershed to sequester $CO_2$. $\kappa_{stream}$ is the amount of $CO_2$ emitted or sequestered calculated from $[\Sigma^+]_{total}$ as described above, normalized by $[\Sigma^+]_{total}$ (meq/l):

$$\kappa_{stream} = -\frac{\Delta CO_2}{[\Sigma^+]_{total}}, \tag{S9}$$

A watershed with no carbonate nor pyrite has the highest capacity to sequester $CO_2$ and $\kappa_{stream}$ equals -0.5. The negative sign is used so that a negative $\kappa_{stream}$ represents sequestration (uptake of $CO_2$), and a positive $\kappa_{stream}$ represents release. Substituting from eq. S8 into eq. S9 yields:

$$\kappa_{stream} = -\frac{0.5 \, [\Sigma^+]_{silicate-CO_2} - 0.25 \, [\Sigma^+]_{carbonate-H_2SO_4}}{[\Sigma^+]_{total}}, \tag{S10}$$

We can further expand eq. S10 by substituting eq. S4 for $[\Sigma^+]_{carbonate-H2SO4}$, eq. S7 for $[\Sigma^+]_{silicate-CO2}$, eq. S2 for $[\Sigma^+]_{silicate}$ and eq. S6 for $[\Sigma^+]_{silicate-H2SO4}$

$$\kappa_{stream} = -\frac{0.5\left([\Sigma^+]_{total} - [\Sigma^+]_{carbonate} - \alpha_{shallow}[SO_4^{2-}]_{total}\right) - 0.5 \, \alpha_{deep}[SO_4^{2-}]_{total}}{[\Sigma^+]_{total}}, \tag{S11}$$

This can be rearranged and simplified as:

$$\kappa_{stream} = -\frac{1}{2} + \frac{1}{2}\frac{[\Sigma^+]_{carbonate}}{[\Sigma^+]_{total}} + \frac{1}{2}\frac{[SO_4^{2-}]_{total}}{[\Sigma^+]_{total}}, \tag{S12}$$

We then defined the second term (ratio of carbonate-derived base cations to total base cations in the stream sample) as $\gamma_{stream}$ and the third term (ratio of the proton equivalents of sulfuric acid to the equivalents of base cations in the stream) as $\zeta_{stream}$. Given these definitions, eq. S12 yields eq. 2 from the main text:

$$\kappa_{stream} = \frac{1}{2}(-1 + \gamma_{stream} + \zeta_{stream}) \tag{2}$$

7. SM, line 55-57. If I understand correctly, here you attribute all Ca and Mg to carbonate dissolution. However, it can also come from silicates. In fact, in figure 1 you represent silicates by CaSiO2. Do you simply neglect Ca from silicate weathering?

We do not attribute all Ca and Mg to carbonate weathering. We attribute Ca and Mg in the deep-weathering endmember to carbonates and Ca and Mg in the shallow-weathering to silicates. In particular, we attribute Mg in the shallow-weathering endmember to the dissolution of chlorite. We do recognize that assuming all Ca and Mg in the deep weathering endmember could be over-estimating the carbonate fraction; however, due to the strong correlation between Ca and Mg

concentrations in this endmember, we believe this to be a good assumption. Moreover, we have not observed any Ca-bearing silicate minerals at Shale Hills.

We have updated the text to increase clarity:

"Note that because of the correlation between Ca and Mg in the deep-weathering endmember, we assume that all of the Ca and Mg in this endmember are associated with carbonates; however, there are Ca and Mg associated with silicate minerals in the shallow-weathering endmember. At Shale Hills, all of the Ca is associated with the deep weathering endmember (see Table S1). This is consistent with the fact that no Ca-bearing silicate minerals have been observed at Shale Hills."

Technical corrections

1. SM, line 28. Change "in in" to "in".
2. SM, line 133-134. I think this equation is equation 3 from the main text, not 2.
3. SM, line 148. You refer to Fig. 3C. However, this figure contains nothing related to lag times. I think you mean Fig. 4C.

We thank the reviewer for the technical corrections and have updated the text to reflect these changes.

---

## Author Comment (AC2) · 20 Jan 2021

Response to Reviewer 2.

This study focuses on applying machine learning to endmember mixing analysis of weathering chemistry in subsurface groundwater flow paths. They apply an NMF scheme and train it on syntactic data generated using a multivariate normal distribution of log-transformed stream water chemistries. The NMF is then applied to 3 measured stream water samples to delineate mixing proportions. The study is well presented and written, the SM is seminal to the understanding of the study and holds the key details for the optimization of the NMF. The main finding is within the sensitivity of the reaction to the groundwater flow paths which are unknown, yet they control the concentration relation between the components and therefore they are controlling the overall chemistry. In a way, these flow paths are a spatial localization of the reaction in space and time, due to the seasonal effects, as shown here. I found the paper very interesting, well written, and clear and supports the publication of the study, yet the main missing part that is not discussed here and must be added is a discussion on the how.

We thank the reviewer for their thoughtful summary, reviewing our work, and for their support in publishing our work. Below are reposes pertaining to your questions of "how".

How does the NMF manage to capture the effect of the subsurface groundwater flow paths? What is the additional mechanism that is deciphered by the NMF? The spatial and temporal effect of the subsurface groundwater flow paths must be captured in a meanfield way by the MNF, and this is not clear how it managed to do so and what was the missing mechanism.

NMF determines patterns in datasets. As domain scientists, we then interpret those patterns in the context of our system. For example, at Shale Hills, the patterns derived by NMF show that one endmember of the stream chemistry has high $[Ca^{2+}]/[SO_4^{2-}]$ and $[Mg^{2+}]/[SO_4^{2-}]$ ratios. Using our prior knowledge, we recognize that water that flows deep at Shale Hills predominantly dissolves carbonate (i.e., calcite and ankerite) and pyrite. It is important to know that NMF does not "know" anything about flowpaths, it only derives the patterns that we are interpreting as flowpaths.

In the past, scientists have used their domain knowledge to define endmembers for EMMA and our new approach uses domain knowledge after the endmembers have been derived by NMF. The main idea behind our interpretations is that mineral reaction fronts commonly separate in the subsurface (Brantley et al., 2017); therefore, different flowpaths dissolve different minerals, and patterns of these mineral dissolution reactions can be detected using NMF.

We thank the reviewer for the comments, and we will expand the discussion in sections 3.2 and 3.5, as well as add information for where we think NMF could and could not be helpful. For example, in section 3.2, we will add: "The dissolution of different minerals along these flowpaths lead to patterns in stream chemistry that our NMF model discerns and separates the signals. If mineral reaction fronts are not separated in the subsurface, different flowpaths might not be separated by NMF; however, Gu et al. (2020) shows that separation of reaction fronts is common in shale watersheds."

Reference:

Brantley, S. L., Lebedeva, M. I., Balashov, V. N., Singha, K., Sullivan, P. L., & Stinchcomb, G. (2017). Toward a conceptual model relating chemical reaction fronts to water flow paths in hills. *Geomorphology*, *277*, 100-117.

I agree with referee 1 remarks 5 and 6, do clarify the mathematical components with a mathematical expression.

We agree with the reviewer and we have updated the text to reflect the changes. Please see the response to reviewer 1 for full derivation of the mathematical expressions outlined in remarks 5 and 6.

---

## Author Response (AR1)

Response to Reviewer 1.

General comments:
The paper applies NMF (Non-negative Matrix Factorization), which is a machine learning technique, to EMMA (End-Member Mixing Analysis). They use this to calculate $CO_2$ sequestration in three watersheds. The novelty is the application of NMF to EMMA. In general, the paper is well written. I suggest publication if the comments below are addressed.

We thank the reviewer for providing helpful comments towards improving the paper. We incorporated the comments suggested here, which we believe strengthened the manuscript. Below are the specific responses to each comment.

Specific comments

1. Line 19-20, 44-45 and 412. You talk about a "new machine learning technique". Actually, it is not a new technique. What you do is applying an old technique (machine learning or, more specifically, NMF) to EMMA, which is new.

We agree with the reviewer that the technique is not new, but rather the application of the technique. We have updated the manuscript to reflect these changes.

Specific examples of our edits:
- Lines 172-173 we clarify that NMF has been used in other applications. "NMF is an algorithm that has been used for many applications (e.g., spectral analysis, email surveillance, cluster analysis; Berry et al., 2007) but has only recently been applied to stream chemistry (e.g., Xu and Harman, 2020)."

2. Line 132-134. You say that NMF is unique in that it does not rely on assumptions of endmembers a priori. This is repeated throughout the whole paper (figure 2, line 172, 412 and 428). I think this is not entirely true. For instance, Carrera et al. (2004) calculate endmembers without NMF. Carrera, J., E. Vázquez-Suñé, O. Castillo, and X. Sánchez-Vila (2004), A methodology to compute mixing ratios with uncertain endmembers, Water Resour. Res., 40, W12101, doi: 10.1029/2003WR002263.

We thank the reviewer for the reference. In the revised manuscript we emphasize that this technique is different than traditional inverse methods and acknowledge previous work that has improved EMMA through modeling under-constrained endmembers (including Carrera et al., 2004).

Specific example of our edits:
- Lines 79-82 we discuss prior contributions. "Since the inception of EMMA, many researchers have aimed to improve analysis through a more accurate determination of unknown or under-constrained endmember chemistries (Hooper, 2003; Carrera et al., 2004; Valder et al., 2012). But these efforts all use some a priori determination of

endmembers. Our machine learning model adds to the growing effort to improve EMMA by applying blind source separation."

3. Line 138: You use SO4 as a reference for solute concentrations. To me it would make more sense to use Cl-, instead, because it is not likely involved in chemical reactions. Is there a particular reason for using SO4?

We thank the reviewer for their question. We specifically normalize to sulfate because it is the target analyte that we wish to separate in the stream. We have updated the text to highlight the rationale behind normalizing to sulfate.

Specific example of our edits:
- Lines 145-151 we discuss the normalization. "Here, cell entries of $V$ are molar solute concentration ratios, $[X]/[Y]$, for stream samples. Indicator $n$ refers to the sampling date, $m$ refers to different solutes $X$ (= $Ca^{2+}$, $Mg^{2+}$, $Na^+$, $K^+$, $Cl^-$), and brackets refer to concentrations. $W$ is the $n$ x $p$ matrix whose cell entries are proportions, $\alpha$, for each endmember in each stream sample. Again, $n$ refers to sampling dates, but $p$ is the number of sources of solutes (referred to as endmembers). The proportions refer to the fractions of sulfate in each sample that derive from an individual endmember, where the sum of proportions must equal $1 \pm 0.05$ for each sample. To derive the mixing proportions of sulfate specifically, we set up the NMF approach by normalizing each analyte concentration by sulfate concentration ($Y = SO_4^{2-}$), the target solute."

4. Line 145: You define end members for shallow, moderately shallow, and deep flowpaths. Of course, they may vary in time as you say in line 149. Could this create some bias? For example, end members of deep flowpaths are generally older with water that fell as rainfall earlier than end members of shallow flowpaths. As acid rain varies with time, differences in chemical signature can be affected by the age of the water.

We thank the reviewer for their thoughtful question. Although different flowpaths do have different transit times, we believe that our model is accurately separating acid rain and pyrite-derived sulfate.

Specific examples of evidence in our text:
- Figure 3 and lines 198-217 describe the separation of reaction fronts in the subsurface at Shale Hills. Lines 230-235 relate the NMF derived endmember chemistries to flowpaths based on the subsurface structure.
- Figure 4A and Lines 244-247 describe how our NMF model results compare to sulfur isotope values, which are not included in the model. The $\alpha_{deep}$ values and low $\delta^{34}S$ values shown in Fig. 4A are consistent with our interpretation that we are separating pyrite-derived sulfate and inconsistent with older acid rain.

5. Line 265: Equation 3 and kstream are not clear to me. Where does the -1 come from? I suggest adding an explanation in the SM like you have done for krock.

We thank the reviewer for this suggestion. We have updated the supplemental to include a derivation of $\kappa_{stream}$ like we did $\kappa_{rock}$.

Specific example of our edits:
- SM Section 2.2, eqs. S11-S14:

"Next, we will derive $\kappa_{stream}$, the modern $CO_2$ sequestration coefficient. In general, both $\kappa_{stream}$ and $\kappa_{rock}$ (see SM 2.3) are used as ways to note the extent that weathering in a watershed is sequestering or releasing $CO_2$. $\kappa_{stream}$ is the amount of $CO_2$ emitted or sequestered calculated from $[\Sigma^+]_{total}$ as described above, normalized by $[\Sigma^+]_{total}$ (meq/l):

$$\kappa_{stream} = -\frac{\Delta CO_2}{[\Sigma^+]_{total}}, \qquad (S11)$$

The negative sign is used so that a negative $\kappa_{stream}$ represents sequestration (uptake of $CO_2$), and a positive $\kappa_{stream}$ represents release. From eq. S11 it is apparent that the $CO_2$ emitted or sequestered equals the product, $\kappa_{stream} [\Sigma^+]_{total}$, with the appropriate sign. Total dissolved base cations in a stream draining a watershed with no carbonate nor pyrite are attributed here entirely as $CO_2$-weathering: this watershed demonstrates the highest capacity to sequester $CO_2$ and $\kappa_{stream}$ equals -0.5. Substituting from eq. S10 into eq. S11 yields:

$$\kappa_{stream} = -\frac{0.5\,[\Sigma^+]_{silicate-CO_2} - 0.25\,[\Sigma^+]_{carbonate-H_2SO_4}}{[\Sigma^+]_{total}}, \qquad (S12)$$

We can further expand eq. S12 by substituting eq. S6 for $[\Sigma^+]_{carbonate-H2SO4}$ , eq. S9 for $[\Sigma^+]_{silicate-CO2}$ , eq. S4 for $[\Sigma^+]_{silicate}$ and eq. S8 for $[\Sigma^+]_{silicate-H2SO4}$

$$\kappa_{stream} = -\frac{0.5\left([\Sigma^+]_{total} - [\Sigma^+]_{carbonate} - 2\alpha_{shallow}[SO_4^{2-}]_{total}\right) - \alpha_{deep}[SO_4^{2-}]_{total}}{[\Sigma^+]_{total}}, \qquad (S13)$$

This can be rearranged and simplified as:

$$\kappa_{stream} = -\frac{1}{2} + \frac{1}{2}\frac{[\Sigma^+]_{carbonate}}{[\Sigma^+]_{total}} + \frac{[SO_4^{2-}]_{total}}{[\Sigma^+]_{total}}, \qquad (S14)$$

We then define the second term (ratio of carbonate-derived base cations to total base cations in the stream sample) as $\gamma_{stream}$ and the third term (ratio of the sulfate equivalents (from sulfuric acid) to the equivalents of base cations in the stream) as $\zeta_{stream}$. Note that to obtain the sulfate equivalents, we multiply $[SO_4^{2-}]_{total}$ by 2, resulting in the third term equal to $0.5\zeta_{stream}$. Given these definitions, eq. S14 yields eq. 2 from the main text:

$$\kappa_{stream} = \frac{1}{2}(-1 + \gamma_{stream} + \zeta_{stream})"$$

6. SM, section 2.2. I find this section very hard to follow. Actually, you describe mathematical equations by using text. I think you can make it more readable, if you put the equations as well.

We agree with the reviewer that adding equations to describe the calculation increases the clarity. We have updated SM section 2.2 to reflect these changes:

Specific example of our edits:
- SM Section 2.2, eqs. S3-S10:

"Here we calculate the inferred $CO_2$ release or sequestration resulting from weathering as recorded in the sum of all base cation concentrations (meq/l) in each stream sample, $[\Sigma^+]_{total}$:

$$[\Sigma^+]_{total} = 2[Ca^{2+}]_{total} + 2[Mg^{2+}]_{total} + [Na^+]_{total} + [K^+]_{total}, \qquad (S3)$$

Here, we use the modeled base cation concentrations from NMF in eq. S3, and we use the uncertainty in the modeled concentrations for the error in $[\Sigma^+]_{total}$. To calculate the inferred $CO_2$ release or sequestration resulting from weathering, we use the results of NMF, as described below, to identify the extents of 4 weathering reactions recorded in each stream sample: 1) $CO_2$-driven weathering ($CO_2$-weathering) of silicates, 2) $H_2SO_4$-driven weathering ($H_2SO_4$-weathering) of silicates, 3) $CO_2$-weathering of carbonates, and 4) $H_2SO_4$-weathering of carbonates. We note these four quantities respectively as 1) $[\Sigma^+]_{carbonate-CO_2}$; 2) $[\Sigma^+]_{silicate-H_2SO_4}$; 3) $[\Sigma^+]_{silicate-CO_2}$; 4) $[\Sigma^+]_{carbonate-H_2SO_4}$. These are the four unknowns we seek to calculate for SH and ER, as described below.

Based on the high proton and low metal concentrations of the measured rain chemistry, the rain contributes negligibly to the base cation concentrations of the study streams; therefore, we apportioned all the base cations to weathering reactions. First, we note that the meq/l of cations derived from carbonate minerals, $[\Sigma^+]_{carbonate}$, equal $[\Sigma^+]_{carbonate-CO_2} + [\Sigma^+]_{carbonate-H_2SO_4}$. Likewise, the meq/l of cations derived from silicate minerals, $[\Sigma^+]_{silicates}$ equal $[\Sigma^+]_{silicate-H_2SO_4} + [\Sigma^+]_{silicate-CO_2}$. The summation of silicate-cations ($[\Sigma^+]_{silicate}$) is the difference between the summation of total cations ($[\Sigma^+]_{total}$) and that of carbonate-derived cations ($[\Sigma^+]_{carbonate}$):

$$[\Sigma^+]_{silicate} = [\Sigma^+]_{total} - [\Sigma^+]_{carbonate}, \qquad (S4)$$

We use a few field observations to complete the calculations for SH and ER, as explained in the main text. First, carbonate minerals only dissolve in water flowing along the deep path because carbonates have been depleted from shallow depths. Second, although some chlorite dissolves into water flowing along the deep path, the release of Mg at depth is insignificant compared to Mg released from carbonate. So we ignore shallow dissolution of carbonates and deep dissolution of silicates in both SH and ER.

With these observations, we can write:

$$[\Sigma^+]_{carbonate} = 2\alpha_{deep}[SO_4^{2-}]_{total}\left(\left(\frac{[Ca^{2+}]}{[SO_4^{2-}]}\right)_{deep} + \left(\frac{[Mg^{2+}]}{[SO_4^{2-}]}\right)_{deep}\right), \tag{S5}$$

Here, $\alpha_{deep}$ is the proportion determined through NMF of sulfate in a given water sample that was derived from reactions along the deep flowpath, $[SO_4^{2-}]_{total}$ is the total concentration of sulfate in the stream water sample under consideration, $([Ca^{2+}]/[SO_4^{2-}])_{deep}$ and $([Mg^{2+}]/[SO_4^{2-}])_{deep}$ are the model-derived ratios of $[Ca^{2+}]$ and $[Mg^{2+}]$ to $[SO_4^{2-}]$, respectively, that characterize the deep flowpath endmember for that sample.

Remembering that Mg release from chlorite dissolution at depth is insignificant compared to Mg from carbonates, all of the generated sulfate in the deep weathering endmember is balanced by cations from dissolved carbonate minerals:

$$[\Sigma^+]_{carbonate-H_2SO_4} = 4\alpha_{deep}[SO_4^{2-}]_{total}, \tag{S6}$$

(We multiply the concentration of deep sulfate by 4 because 4 eq of cations are released per mol of sulfate, noting that $[\Sigma^+]$ is in eq/L and $[SO_4^{2-}]$ is in mol/L). Any carbonate-derived base cations that are in excess of what could have been produced by pyrite-derived sulfuric acid are attributed to $CO_2$-weathering of carbonates:

$$[\Sigma^+]_{carbonate-CO_2} = [\Sigma^+]_{carbonate} - [\Sigma^+]_{carbonate-H_2SO_4}, \tag{S7}$$

Remembering that no carbonates dissolve into water flowing along the shallow path, then similar arguments for the shallow flowpath yield:

$$[\Sigma^+]_{silicate-H_2SO_4} = 2\alpha_{shallow}[SO_4^{2-}]_{total}, \tag{S8}$$
$$[\Sigma^+]_{silicate-CO_2} = [\Sigma^+]_{silicate} - [\Sigma^+]_{silicate-H_2SO_4}, \tag{S9}$$

From these equations, values for the four unknowns can be calculated for SH and ER. A similar approach was taken for HB except that no carbonate minerals were present, and only two unknowns were determined ($[\Sigma^+]_{silicate-H_2SO_4}$, $[\Sigma^+]_{silicate-CO_2}$).

With respect to the atmosphere considered over the long-term ($10^5$-$10^6$ yr), $H_2SO_4$-weathering of silicates and $CO_2$-weathering of carbonates are $CO_2$ neutral, while $CO_2$-weathering of silicates sequesters $CO_2$ and $H_2SO_4$-weathering of carbonates releases $CO_2$ (Fig. 1). As seen in Figure 1, per mole of $CaSiO_3$ or $CaCO_3$ weathered, $CO_2$-weathering of silicates sequesters 1 mol of $CO_2$ and $H_2SO_4$-weathering of carbonates releases 0.5 moles of $CO_2$. In terms of $[\Sigma^+]_{total}$, $CO_2$-weathering of silicates sequesters 0.5 moles of $CO_2$ per base cation equivalent released into solution and $H_2SO_4$-weathering of carbonates releases 0.25 moles of $CO_2$ per base cation equivalent released into solution (Fig. 1; Reactions 2, 3, 6, and 7). For a given water sample, the cation concentrations record the extent of dissolution of carbonate and silicates, as long as the contribution of these base cations from acid rain is minimal. (For simplicity, we do not correct $[\Sigma^+]$ for rain chemistry but see SM Section 4). Therefore, the uptake or release of $CO_2$, $\Delta CO_2$, can be calculated for any given stream water sample:

$$\Delta CO_2 = 0.5 \, [\Sigma^+]_{silicate-CO_2} - 0.25 \, [\Sigma^+]_{carbonate-H_2SO_4}. \tag{S10}"$$

7. SM, line 55-57. If I understand correctly, here you attribute all Ca and Mg to carbonate dissolution. However, it can also come from silicates. In fact, in figure 1 you represent silicates by CaSiO2. Do you simply neglect Ca from silicate weathering?

We do not attribute all Ca and Mg to carbonate dissolution, but rather all Ca and Mg in the deep weathering flowpath to carbonate dissolution. Ca and Mg in the shallow-weathering flowpath are thus attributed to silicate dissolution.

Specific example of our edits:
- Eq. S5 shows that carbonate cations are only derived from Ca and Mg in the deep weathering flowpath: $[\Sigma^+]_{carbonate} = 2\alpha_{deep}[SO_4^{2-}]_{total}\left(\left(\frac{[Ca^{2+}]}{[SO_4^{2-}]}\right)_{deep} + \left(\frac{[Mg^{2+}]}{[SO_4^{2-}]}\right)_{deep}\right)$

Technical corrections

1. SM, line 28. Change "in in" to "in".
   - SM line 31 was revised to correct this mistake.
2. SM, line 133-134. I think this equation is equation 3 from the main text, not 2.
   - SM line 233 was revised to correct this mistake.
3. SM, line 148. You refer to Fig. 3C. However, this figure contains nothing related to lag times. I think you mean Fig. 4C.
   - SM line 249 was revised to correct this mistake.

Response to Reviewer 2.

This study focuses on applying machine learning to endmember mixing analysis of weathering chemistry in subsurface groundwater flow paths. They apply an NMF scheme and train it on syntactic data generated using a multivariate normal distribution of log-transformed stream water chemistries. The NMF is then applied to 3 measured stream water samples to delineate mixing proportions. The study is well presented and written, the SM is seminal to the understanding of the study and holds the key details for the optimization of the NMF. The main finding is within the sensitivity of the reaction to the groundwater flow paths which are unknown, yet they control the concentration relation between the components and therefore they are controlling the overall chemistry. In a way, these flow paths are a spatial localization of the reaction in space and time, due to the seasonal effects, as shown here. I found the paper very interesting, well written, and clear and supports the publication of the study, yet the main missing part that is not discussed here and must be added is a discussion on the how.

We thank the reviewer for their thoughtful summary, reviewing our work, and for their support in publishing our work. Below are reposes pertaining to your questions of "how".

How does the NMF manage to capture the effect of the subsurface groundwater flow paths? What is the additional mechanism that is deciphered by the NMF? The spatial and temporal

effect of the subsurface groundwater flow paths must be captured in a meanfield way by the MNF, and this is not clear how it managed to do so and what was the missing mechanism.

NMF determines patterns in datasets. As domain scientists, we then interpret those patterns in the context of our system. We describe in detail in section 3.2 how we interpret the endmembers and relate them to flowpath and sub-surface structure. In a practical sense, water that flows along a flowpath dissolved minerals that are present along that flowpath. Because there is a separation on reaction fronts at our sites, each flowpath interacts with different minerals and, therefore, has a unique chemical signature that can be detected and separated with NMF. We have updated the text to clarify this idea.

Specific example of our edits:
- Lines 259-262 describe the importance of reaction front separation. "The dissolution of different minerals along these flowpaths lead to patterns in stream chemistry that our NMF model discerns and separates. If mineral reaction fronts are not separated in the subsurface, different flowpaths might not be separated by NMF; however, Brantley et al. (2017) and Gu et al. (2020a) have argued that separation of reaction fronts is common."

I agree with referee 1 remarks 5 and 6, do clarify the mathematical components with a mathematical expression.

Specific example of our edits:
- See SM section 2.2 eqs. S3-S14